

# Aerosol optical depth retrieval from the EarthCARE multi-spectral imager: the M-AOT product

Nicole Docter[1], Rene Preusker[1], Florian Filipitsch[1,2], Lena Kritten[1], Franziska Schmidt[1,3], and Jürgen Fischer[1]

[1]Freie Universität Berlin (FUB), Institute for Space Science, Berlin, Germany
[2]now at: Deutscher Wetterdienst (DWD), Meteorological Observatory Lindenberg (MOL-RAO), Tauche, Germany
[3]now at: Freie Universität Berlin (FUB), Institute for Meteorology, Berlin, Germany

**Correspondence:** Nicole Docter (nicole.docter@fu-berlin.de)

**Abstract.** The Earth Explorer mission Earth Clouds, Aerosols and Radiation Explorer (EarthCARE) will not only provide profile information on aerosols but will also deliver a horizontal context to it through measurements by its Multi-Spectral Imager (MSI). The columnar aerosol product relying on these passive signals is called M-AOT. Its main parameters are aerosol optical thickness (AOT) at 670 nm over ocean and, where possible land, and at 865 nm over ocean. Here, the algorithm and

assumptions behind it are presented. Further, first examples of product parameters are given based on applying the algorithm to simulated EarthCARE test data and Moderate Resolution Imaging Spectroradiometer (MODIS) Level-1 data. Comparisons to input fields used for simulations, to the official MODIS aerosol product, AErosol RObotic NETwork (AERONET) and to Maritime Aerosol Network (MAN) show an overall reasonable agreement. Over ocean correlations are 0.98 (simulated scenes), 0.96 (compared to MYD04) and 0.9 (compared to MAN). Over land correlations are 0.62 (simulated scenes), 0.87 (compared

to MYD04) and 0.77 (compared to AERONET). A concluding discussion will focus on future improvements necessary and envisioned to enhance the product.

## 1 Introduction

The Earth Clouds, Aerosols and Radiation Explorer (EarthCARE) aims to improve the understanding of interaction between clouds, aerosols and radiation (Illingworth et al., 2015; Wehr et al., 2022). Two active and one passive remote sensing instru-

ment are carried on board the spacecraft in order to monitor the horizontal and vertical distribution of clouds and aerosols simultaneously from one platform. Aerosols have a special role in the overall context since they do not only directly interact with radiation through scattering and absorption but are also indirectly affecting radiative forcing through their influence on cloud optical properties, e.g. by acting as cloud condensation and ice nuclei. Uncertainties of aerosol radiative forcing are present due to an incomplete knowledge of aerosol optical properties and its spatial and temporal variability. Hence, concern-

ing aerosols, EarthCARE was designed in such a way that vertical aerosol property information based on ATmospheric LIDar (ATLID) measurements can be put into a horizontal context based on information provided by Multi-Spectral Imager (MSI) measurements (Wehr et al., 2022).



While the configuration of ATLID allows for more scientifically novel space-based aerosol parameters to be retrieved (Donovan et al., 2022b), the addition of MSI observations will further allow to align newly gained knowledge about the aerosol composition to well established heritage and ongoing columnar aerosol products via EarthCARE's own imager measurements. The presented algorithm and its accompanied product provide conventional imager aerosol information within the retrieval chain (Eisinger et al., 2022). They are both called M-AOT, which is short for MSI-Aerosol Optical Thickness.

The algorithm is building on heritage knowledge, assumptions and methods to derive aerosol optical thickness (AOT) based on imager measurements adjusted to MSI specifications. In particular, it is taken advantage of the brightening of the top-of-atmosphere (TOA) signal in presence of aerosol scattering over a relatively dark surface. While the surface contribution over open ocean outside of glint regions is negligible, in particular in the near-infrared (NIR) and shortwave-infrared (SWIR), over land surfaces, it needs to be taken into account. First imager based aerosol retrievals have been applied over ocean surfaces using Advanced Very High Resolution Radiometer (AVHRR, e.g. Husar et al., 1997; Higurashi and Nakajima, 1999; Mishchenko et al., 1999; Geogdzhayev et al., 2002; Ignatov et al., 2004), Moderate Resolution Imaging Spectroradiometer (MODIS, Tanré et al., 1997), or Meteosat (Moulin et al., 1997). One of the most commonly known algorithms used for aerosol optical property retrievals over land and ocean is the dark target (DT) approach (e.g. Remer et al., 2020). It has been applied to, e.g. MODIS (e.g. Kaufman et al., 1997b, a; Remer et al., 2005; Levy et al., 2007a, b, 2009, 2013). Other imager based aerosol products rely on measurements of, for example, AVHRR (Hsu et al., 2017), Visible Infrared Imaging Radiometer Suite (VIIRS, Jackson et al., 2013; Levy et al., 2015; Patadia et al., 2018; Sawyer et al., 2020), Spinning Enhanced Visible and InfraRed Imager (SEVIRI Wagner et al., 2010; Luffarelli and Govaerts, 2019; Ceamanos et al., 2023), (Advanced) Along-Track Scanning Radiometer ((A)ATSR, e.g. Veefkind et al., 1999; North, 2002; Grey et al., 2006; Curier et al., 2009; Thomas et al., 2009; Bevan et al., 2012; Kolmonen et al., 2016), Medium Resolution Imaging Spectrometer (MERIS, e.g. von Hoyningen-Huene et al., 2006; Katsev et al., 2009; Mei et al., 2017) and successor Ocean and Land Colour Instrument (OLCI, Mei et al., 2018; Chen et al., 2022). This list is far from exhausted but is intended to demonstrate the wide availability of passive, columnar aerosol algorithms and knowledge available in literature. Also, more sophisticated aerosol property retrieval products are available in recent times using instruments that provide additional pieces of measurement information, e.g. multi-angle and polarized measurements, which will not be available from MSI. Instruments providing such measurements and, hence, additional aerosol property information besides AOT are e.g. Polarization and Directionality of the Earth's Reflectances (POLDER, e.g. Dubovik et al., 2011) or, in the future, the EPS-SG Multi-Viewing Multi-Channel Multi-Polarisation Imaging (3MI) instrument.

For a heritage instrument like MSI, the optimal estimation (OE) technique (Rodgers, 2000) is one attempt to overcome the problem of separating the aerosol and the surface signal by using prior knowledge in order to constrain the result. This method is used in other remote sensing applications (e.g. Govaerts et al., 2010; Sayer et al., 2012; Jeong et al., 2016). A notable advantage of this technique is the traceability of uncertainties.

Relying on all this past research, MSI's four visible to short-wave infrared bands at 670 nm (VIS), 865 nm (NIR), 1650 nm (SWIR-1) and 2200 nm (SWIR-2) are used to derive AOT over ocean at 670 nm and 865 nm as well as, where possible, at 670 nm over vegetated land surfaces on the native spatial resolution of 500 m x 500 m for its swath width of 150 km. In





addition, M-AOT also provides information about the Ångström parameter between 670 nm and 865 nm for each ocean pixel for which AOT was retrieved successfully.

This paper is structured as follows. The operational Level-2 M-AOT algorithm including details on the used forward model and retrieval technique are introduced and its limitations are described in section 2. First example M-AOT products and verification results are presented using simulated EarthCARE test data (Donovan et al., 2022a) and MODIS data within the M-AOT algorithm in sections 3 and 4, respectively. Finally, results will be discussed in section 5.

## 2 M-AOT algorithm description

The M-AOT algorithm (Fig. 1) relies on measurements in the four MSI channels from the visible to short-wave infrared, the cloud mask (Hünerbein et al., 2022) and additionally needed atmospheric parameters provided by X-MET (Eisinger et al., 2022). After correcting the measured signal for residual absorption by gases in the window channels of MSI (sec. 2.2), separate methods for the retrieval above ocean (sec. 2.3.3) and land surfaces (sec. 2.3.4) are applied. For both retrieval branches the OE (sec. 2.3.2) inversion technique is applied. Land and water forward operators (sec. 2.3.1) are making use of pre-calculated Look-up Tables (LUTs) that describe the coupled surface-atmosphere signal. They are based on radiative transfer (RT) simulations.

### 2.1 Initialization and valid pixel identification

The aerosol retrieval core of M-AOT expects normalized TOA radiances $L_\lambda^N$ as input. These are defined as measured MSI spectral radiances $L_\lambda$ normalized by the corresponding spectral inband solar irradiance $E_{0,\lambda}$:

$$L_\lambda^N(\theta_s, \theta_v, \varphi) = \frac{L_\lambda(\theta_s, \theta_v, \varphi)}{E_{0,\lambda}} \tag{1}$$

The normalized TOA radiance for each pixel depends on the solar zenith angle $\theta_s$, the viewing zenith angle $\theta_v$ and the relative azimuth angle $\varphi$, defined by the difference between sun ($\varphi_s$) and instrument viewing azimuth ($\varphi_v$) angles. The angle dependency will, from now on, be omitted from equations but stays implicitly conserved. All these quantities are available from the MSI Level-1 product M-RGR together with the corresponding pixel elevation, which is based on a digital elevation model (DEM).

Level-2 MSI cloud-mask and surface classification provided by the M-CM product (Hünerbein et al., 2022) are used to identify suitable pixel. Only pixel that are indicated as "confident clear" by the cloud mask are used in subsequent steps. Additionally, neighbouring pixel of a cloud are flagged as well during that step in order to avoid sub-pixel clouds or cloud shadow effects. Currently, the size of this cloud-buffer is 3 pixel. This value is based on algorithm testing with simulated EarthCARE test scenes and MODIS inputs.

After the flagging of clouds and cloud edges, the pixel identification will differentiate between ocean and land pixel that are suitable for DT-like retrieval approaches. Therefore, all ocean pixel that are contaminated by sun glint or sea ice and all land



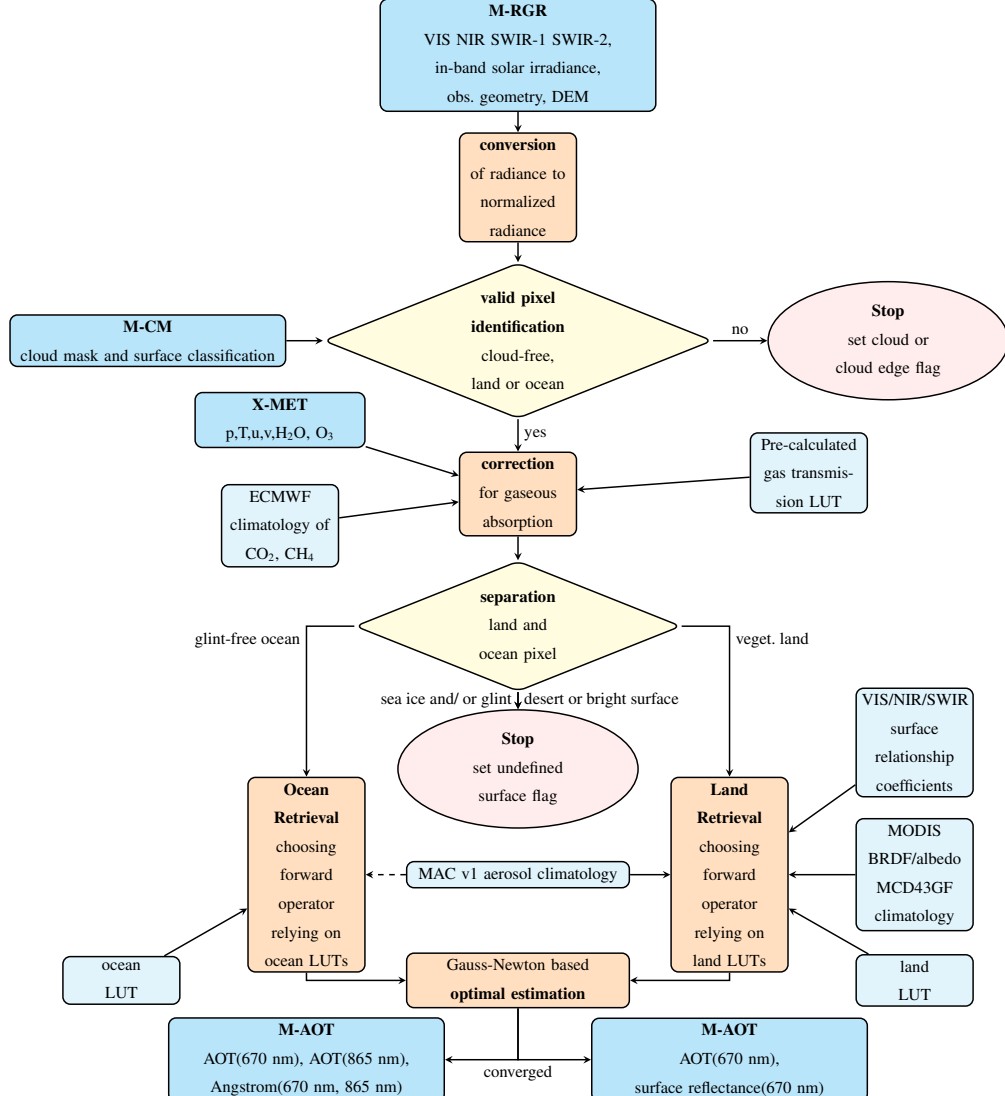

**Figure 1.** M-AOT algorithm flow chart. Blue boxes indicate algorithm inputs, where darker blue colors show EarthCARE chaining products and lighter shaded blues indicate auxiliary inputs. Orange boxes highlight processing steps and yellow boxes indicate decisions that might lead to a stop of an individual AOT retrieval for a pixel.

pixel that are not vegetated land pixel or represent desert pixel or are indicated to be covered by snow will be excluded from any subsequent steps. They all will be flagged as undefined surface pixel pertaining to the scope of M-AOT.





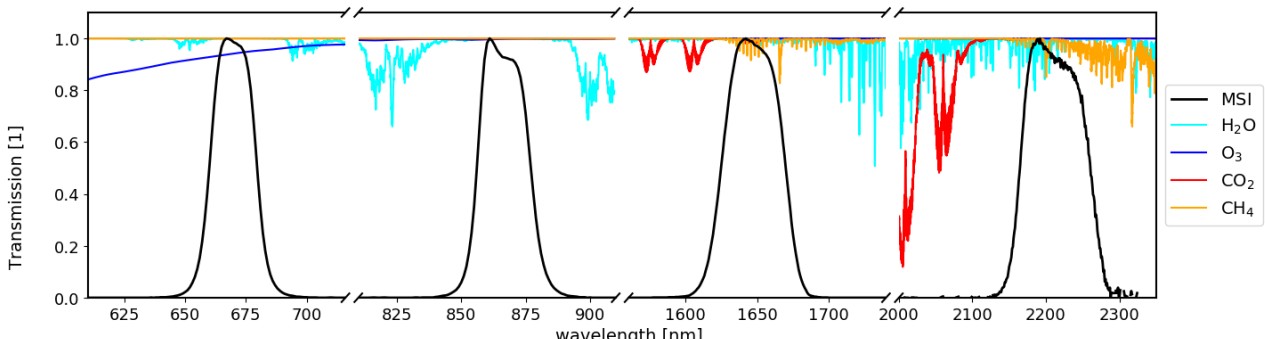

**Figure 2.** Atmospheric gas transmisson of water vapour, ozone, carbon dioxide and methane and MSI nadir response functions of VIS, NIR, SWIR-1, SWIR-2.

## 2.2 Correction for atmospheric gas absorption and accounting for Rayleigh optical thickness

Before the signal is used to determine AOT, the normalized TOA radiance is corrected for absorption by atmospheric gases through the division by atmospheric gas transmission $T_c$:

$$L_{c,\lambda}^N = \frac{L_\lambda^N}{T_c} = \frac{L_\lambda^N}{\prod_{g=1}^n T_g}, \tag{2}$$

where $L_{c,\lambda}^N$ is the gas corrected, normalized TOA radiance and $T_g$ is the transmission associated to an individual gas. This approach is valid as long as the interaction between scattering and gaseous absorption is weak, which is true for the four MSI

bands in the visible to shortwave-infrared.

According to Fig. 2, ozone (e.g. VIS), water vapour (e.g. NIR, SWIR-2), carbon dioxide (e.g. SWIR-1) and methane (e.g. SWIR-1, SWIR-2) need to be considered in the correction process for the VIS, NIR and SWIR-1 and SWIR-2 channels between 670 and 2200 nm. While water vapour and ozone total column amounts are provided by the X-MET product (Eisinger et al., 2022), carbon dioxide and methane amounts are taken from the ECMWF based climatology of Cy43r1 (ECMWF, 2016), which

is based on the MACC reanalysis from 2003-2011 (Inness et al., 2013).

The atmospheric transmission $T_{g,\lambda}$ is available from LUTs for each of these four gases. They have been calculated as a deduction from the Lambert-Beer-Law in advance:

$$T_{g,\lambda} = e^{-\sigma_{g,\lambda} \frac{A}{M_g} VCD(c_g,\mu)} \tag{3}$$

$$T_{g,\lambda} = e^{-\sigma_{g,\lambda} \frac{A}{M_g} c_g \mu} \tag{4}$$

where $\sigma_{g,\lambda}$ is the absorption cross section of the individual gas, given as a function of wavelength, $A$ is the Avogadro constant, $M_g$ is the molar mass of the individual gas, $VCD(c_g,\mu)$ is the vertical column depth in [kg/m$^2$], given as a function of total column gas amount $c_g$ and path length or airmass factor $\mu = (1/cos(\theta_s) + 1/cos(\theta_v))$. These pre-calculated transmission LUTs rely on high resolution absorption cross sections taken from CKDMIP (Correlated K-Distribution Model Intercomparison Project, Hogan and Matricardi (2020)). Gas concentrations, temperature and pressure as present for typical standard





atmospheres (Anderson et al., 1986) have been used to extract typical absorption cross sections from this data base which then have been used to calculate highly resolved transmissions within each bands spectral range. These then have been convolved with the spectral response function corresponding to the nadir across track pixel for each MSI band and have been stored in LUTs. The latter has been chosen in such a way that it covers the global range of gas concentrations for all seasons and possible MSI viewing geometries.

The MSI signal is not corrected for the Rayleigh contribution but rather the Rayleigh optical thickness is explicitly included in the forward simulations. It has been calculated following Bodhaine et al. (1999):

$$\tau_R(\lambda) = \sigma_\lambda^{air} \frac{PA}{(15.0556 \cdot \mathrm{CO}_2 + 28.9595)g}, \tag{5}$$

where $\sigma_\lambda^{air}$ is the scattering cross section of air, $P$ is the surface pressure, $A$ is the Avogadro constant and $g$ is the acceleration of gravity. The $\mathrm{CO}_2$ concentration is assumed to be $0.04\,\mathrm{ppv}$. The surface pressure itself is used as dimension of M-AOT LUTs

in order to parameterize the varying Rayleigh optical thickness. Additionally, to ensure the best possible estimate of the surface pressure for the aerosol retrieval, the provided surface pressure of X-MET is corrected for the height difference between the ECMWF model and the real elevation obtained from the DEM that is contained in the M-RGR product.

## 2.3   Retrieval method

The aerosol retrieval above ocean and land surfaces is based on an OE approach. The scheme minimizes differences of forward

modelled $L_{m,\lambda}^N$ (see sec. 2.3.1) and measured normalized TOA radiance $L_{c,\lambda}^N$ by iteratively varying the AOT value as part of the retrieval state. Measurement and a priori uncertainties are taken into account during that process.

### 2.3.1   Forward model

The forward model relies on pre-generated radiative transfer simulations stored in LUTs. They have been carried out using the matrix-operator model MOMO (Fell and Fischer, 2001; Hollstein and Fischer, 2012) for the combined ocean-atmosphere-land

system. An atmosphere that includes only molecular as well as aerosol scattering and absorption but no gas absorption, is considered. Since MSI channels are only affected to a minor degree, a simple gas correction of the measurements is sufficient as described in section 2.2. A Lambertian surface reflector is assumed for simulations over land. The ocean surface is parameterized following Cox and Munk (1954) for varying wind speeds in order to account for the sea surface roughness. The water body is described in such a way that it is sufficient for open ocean but not for coastal waters. Hence, water LUTs are only valid

for the former.

The LUT sets contain normalized TOA radiances $L_{m,\lambda}^N$ (see Tab. 1) in dependency of viewing geometry (relative azimuth angle $\varphi$, sun zenith angle $\theta_s$, viewing zenith angle $\theta_v$, surface pressure $P$, aerosol optical thickness $\tau_{A,\lambda}$, aerosol composition number $n_A$ and surface reflectance $\rho$ or wind speed $w$ for land and ocean, respectively.

LUT for land surfaces:

$$L_{m,\lambda}^N = f(\varphi, \theta_s, \theta_v, \tau_{A,\lambda}, \rho, P, n_A) \tag{6}$$



**Table 1.** Summary of M-AOT aerosol Look-Up Table dimensions. A linear interpolation is performed in all dimensions but $n_A$.

| Symbol | LUT parameter | sampling point values |
|---|---|---|
| $\varphi$ | relative azimuth angle [°] | 0, 12, 24, 36, 48, 60, 72, 84, 96, 108, 120, 132, 144, 156, 168, 180 |
| $\theta_s$ | sun zenith angle [°] | 0, 1.8, 3.4, 4.9, 6.4, 7.9, 9.4, 10.9, 12.4, 13.9, 15.4, 16.9, 18.5, 21.5, 24.5, 27.5, 30.5, 33.5, 36.5, 39.5, 42.6, 45.6, 48.6, 51.6, 54.6, 57.6, 60.6, 63.6, 66.7, 69.7, 72.7, 75.7, 78.7 |
| $\theta_v$ | viewing zenith angle [°] | 0, 1.8, 3.4, 4.9, 6.4, 7.9, 9.4, 10.9, 12.4, 13.9, 15.4, 16.9, 18.5 |
| $\tau_A$ | AOT(550 nm) [1] | 0.001, 0.01, 0.05, 0.1, 0.3, 0.6, 1.0, 1.5, 2.0, 3.0 |
| $\rho$ | surface reflectance [1] | $0.001^1, 0.01^1, 0.05^{1,2}, 0.1^{1,2}, 0.2^{1,2}, 0.3^{1,2}, 0.5^{1,2}, 0.7^2, 0.9^2$ or |
| $w$ | wind speed [$m\,s^{-1}$] | 2, 5, 9, 13, 17 |
| $P$ | surface pressure [$hPa$] | 800, 850, 900, 950, 1000, 1050, 1100 |
| $n_A$ | number of aerosol component mixing ratio cases, short: composition [1] | 1-25 (for corresponding ratios see Tab. 2) |

1: for NIR only, 2: for VIS, SWIR-1 and SWIR-2

For water surfaces:

$$L_{m,\lambda}^N = f(\varphi, \theta_s, \theta_v, \tau_{A,\lambda}, w, P, n_A) \tag{7}$$

The aerosol optical properties needed for the radiative transfer simulations are AOT, the aerosol scattering phase function $P_A(\Theta)$ and single scattering albedo $\omega_0(\lambda)$ for a given aerosol type, where $\Theta$ is the scattering angle. In order to ensure con-
sistency and comparability between EarthCARE Level-2 products, microphysical properties (size distribution and complex refractive index) of the four HETEAC (Wandinger et al., 2016, 2022) types sea salt, non-spherical dust, fine mode weakly and strongly absorbing aerosol have been used. The optical properties, e.g. phase function, cross section extinction and single scattering albedo, have been generated from these microphysical data. Further, a fixed vertical profile for each of the four basic HETEAC aerosol types is assumed. The aerosol layer height is set according to the aerosol-cci approach (Holzer-Popp et al.,
2013), which assumes it to be at 0-1 km for salt, 2-4 km for dust and 0-2 km for fine mode aerosols. The four basic aerosol types are combined by mixing their individual AOT to 25 predefined compositions for which radiative transfer simulations have been carried out (see Tab. 2).

The standard forward operator used in M-AOT consists of two steps: an n-dimensional interpolation in these pre-calculated LUTs and a subsequent inverse distance weighting. The linear interpolation in the LUTs is conducted for each individual
aerosol composition. Hence, for one pixel and one band, there will be 25 TOA normalized radiance values remaining after that step, which are corresponding to the 25 aerosol compositions. Afterwards, inverse distance weighting is applied for each band individually in order to obtain the normalized TOA radiance for any composition and as selected beforehand for an individual pixel (for details, see sec. 2.3.3, 2.3.4).





**Table 2.** In detail description of pre-defined aerosol optical thickness based component mixing ratios of the individual HETEAC types to their sum of 1 as used for M-AOT aerosol Look-Up Tables

| Composition number ($n_A$) | Sea salt | Dust | Fine mode less absorbing | Fine mode strong absorbing |
|---|---|---|---|---|
| 1 | 0.00 | 0.00 | 0.00 | 1.00 |
| 2 | 0.00 | 0.00 | 1.00 | 0.00 |
| 3 | 0.00 | 0.00 | 0.50 | 0.50 |
| 4 | 0.11 | 0.12 | 0.72 | 0.05 |
| 5 | 0.08 | 0.24 | 0.54 | 0.14 |
| 6 | 0.01 | 0.33 | 0.33 | 0.33 |
| 7 | 0.33 | 0.01 | 0.33 | 0.33 |
| 8 | 0.26 | 0.13 | 0.58 | 0.03 |
| 9 | 0.41 | 0.08 | 0.49 | 0.02 |
| 10 | 0.00 | 0.50 | 0.00 | 0.50 |
| 11 | 0.00 | 0.50 | 0.50 | 0.00 |
| 12 | 0.50 | 0.00 | 0.00 | 0.50 |
| 13 | 0.50 | 0.00 | 0.50 | 0.00 |
| 14 | 0.25 | 0.25 | 0.25 | 0.25 |
| 15 | 0.07 | 0.46 | 0.38 | 0.09 |
| 16 | 0.30 | 0.26 | 0.38 | 0.06 |
| 17 | 0.55 | 0.06 | 0.37 | 0.02 |
| 18 | 0.33 | 0.33 | 0.01 | 0.33 |
| 19 | 0.33 | 0.33 | 0.33 | 0.01 |
| 20 | 0.68 | 0.04 | 0.26 | 0.02 |
| 21 | 0.05 | 0.69 | 0.20 | 0.06 |
| 22 | 0.85 | 0.01 | 0.13 | 0.01 |
| 23 | 0.00 | 1.00 | 0.00 | 0.00 |
| 24 | 1.00 | 0.00 | 0.00 | 0.00 |
| 25 | 0.50 | 0.50 | 0.00 | 0.00 |

### 2.3.2 Optimal estimation

The inverse problem is solved by applying the OE technique as described in (Rodgers, 2000). It determines a state $x$ given observations $y$ and a priori knowledge of the state $x_a$ based on the Bayesian theory taking uncertainties of both into account.





Here, the measurement vector consists of the four MSI normalized TOA radiances $L^N_{c,\lambda}$ that have been corrected for gaseous absorption. The state vector consists of AOT $\tau_A(550\,nm)$ for the aerosol retrieval over vegetated land surfaces and over open ocean. A priori state knowledge is based on the Max-Planck-Institute Aerosol Climatology version 1 (MAC v1) (Kinne et al., 2013) for AOT. The a priori is also used as initial input or so called first guess. The forward operator F($x,b$) also depends on other quantities, $b = [\varphi,\theta_s,\theta_v,P,n_A]$ over land or $b = [\varphi,\theta_s,\theta_v,w,P,n_A]$ over ocean, e.g. viewing geometry, surface pressure and aerosol composition, that influence the measurement (see Tab. 1) but are not part of the retrieved state. The optimal state is found through an iterative approach and by minimizing the cost function:

$$J(\boldsymbol{x}) = (\boldsymbol{x} - \boldsymbol{x}_a)^T S_a^{-1} (\boldsymbol{x} - \boldsymbol{x}_a) + g(\boldsymbol{x})^T S_\epsilon^{-1} g(\boldsymbol{x}) \tag{8}$$

where $g_y = F(\boldsymbol{x},\boldsymbol{b}) - \boldsymbol{y}$, $S_a$ is a priori error co-variance matrix and $S_\epsilon$ is the measurement model error co-variance matrix. In order to find the optimal solution of Eq. 8, the state vector is iteratively updated applying the Gauss-Newton method. There, the state vector for the $(i+1)$-th iteration is given by:

$$\boldsymbol{x}_{i+1} = \boldsymbol{x}_i -$$
$$(S_a^{-1} + K_i^T S_\epsilon^{-1} K_i)^{-1} \tag{9}$$
$$[K_i^T S_\epsilon^{-1} \cdot g(\boldsymbol{x}_i) - S_a^{-1} \cdot (\boldsymbol{x}_a - \boldsymbol{x}_i)]$$

where: $K_i = dF(\boldsymbol{x},\boldsymbol{b})/d\boldsymbol{x}$ is the Jacobian matrix of the $i$-th iteration. As soon as the change in the state compared to the expected uncertainty falls below a certain value, here 0.03 over land and 0.001 over ocean, the iteration loop is exited. The uncertainty of the result is given by:

$$\hat{S} = (S_a^{-1} + K_i^T S_\epsilon^{-1} K_i)^{-1} \tag{10}$$

Hereby, the diagonal elements of this matrix represent the error variances of the state vector elements.

### 2.3.3 Aerosol optical thickness retrieval over ocean

The ocean retrieval branch of the M-AOT algorithm is limited to open ocean surfaces. The needed $10\,m$ wind speeds, are calculated from the $10\,m$ zonal and meridional wind speed of the X-MET product. Since variability in the water leaving radiance is not negligible in coastal regions and other complex waters, they will be flagged out as not suitable for the AOT retrieval. Additionally, areas directly affected by sun glint will not be part of the retrieval above ocean surfaces. Even though MSI is tilted in order to avoid sun glint, it remains apparent for certain observing geometries. Hence, glint affected pixel will be flagged by making use of the glint flag of the M-RGR product and will not be further used.

The MSI instrument alone does only allow for a rough guess of the best fitting aerosol composition to be assumed. However, this is attempted with the intent to achieve an optimal agreement between simulated and measured TOA normalized radiances. It reduces the impact of a wrongly assumed aerosol component ratio mixing in the retrieved AOT. The most likely aerosol composition is searched in an empirical manner following three steps:

1. Spatial averaging of TOA normalized and gas corrected radiance



2. 26 optimal estimation retrievals of AOT for each of these averaged pixel corresponding to 26 pre-defined aerosol compositions

3. Selection of the best fitting composition considering the measurement space

First, MSI normalized radiance values are averaged in order to achieve an operationally required run-time of the M-AOT processor. Therefore, only cloud and glint free ocean pixel are spatially averaged. The pre-launch value, which works best is 10 x 10 pixel. This corresponds to a spatial averaging of $5\,\mathrm{km}$ x $5\,\mathrm{km}$. The value offers the possibility to achieve a balance between spotting small scale patterns but not adding artificially, noisy pixel to the final product.

Secondly, for each averaged pixel, an OE of AOT is applied for each of the 25 theoretically possible compositions, as used for the ocean LUT creation, plus the climatological composition as reported in the MAC v1 climatology. This step delivers not only 26 AOTs for one averaged ocean pixel but also 26 forward modelled normalized radiances in each band.

Thirdly, the best fitting aerosol composition for an averaged pixel is found using three measures describing the spectral behavior and two measures describing the accuracy of the fit. The measures (see Eq. 11-15) are the ratio between forward simulated NIR and SWIR-1 band $r_{\lambda_2,\lambda_3}$, the ratio between SWIR-1 and SWIR-2 band $r_{\lambda_3,\lambda_4}$, the spectral angle $\gamma$ as well as the root mean squared error $rmse$ and the correlation $r_{c,m}$ between simulated and measured normalized radiance:

$$r_{\lambda_2,\lambda_3} = \frac{L^N_{m,\lambda_2}}{L^N_{m,\lambda_3}} \tag{11}$$

$$r_{\lambda_3,\lambda_4} = \frac{L^N_{m,\lambda_3}}{L^N_{m,\lambda_4}} \tag{12}$$

$$\gamma = cos^{-1}\left(\frac{\sum_{i=1}^{n} L_{m,\lambda_i} L_{c,\lambda_i}}{\sqrt{\sum_{i=1}^{n} {L_{m,\lambda_i}}^2}\sqrt{\sum_{i=1}^{n} {L_{c,\lambda_i}}^2}}\right) \tag{13}$$

$$rmse = \sqrt{\frac{\sum_{i=1}^{n} (L^N_{m,\lambda_i} - L^N_{c,\lambda_i})^2}{n}} \tag{14}$$

$$r_{c,m} = \frac{\sum_{i=1}^{n} (L_{m,\lambda_i} - \overline{L}_{m,\lambda_i})(L_{c,\lambda_i} - \overline{L}_{c,\lambda_i})}{\sqrt{\sum_{i=1}^{n} (L_{m,\lambda_i} - \overline{L}_{m,\lambda_i})^2 \sum_{i=1}^{n} (L_{c,\lambda_i} - \overline{L}_{c,\lambda_i})^2}} \tag{15}$$

These measures have been chosen in order to make sure that the spectral behaviour is reasonably represented in the forward simulated measurements and that the differences between forward simulation and measurements are small at the same time. In order to find the best fitting composition during this step, the euclidean distance $e_{n_A}$ between the theoretically best estimates $\boldsymbol{c_e} = [L^N_{c,\lambda_2}/L^N_{c,\lambda_3}, L^N_{c,\lambda_3}/L^N_{c,\lambda_4}, 0, 0, 1]$ and the feature vector $\boldsymbol{f_e} = [r_{\lambda_2,\lambda_3}, r_{\lambda_3,\lambda_4}, rmse, \gamma, r_{c,m}]$ is calculated as follows:

$$e_{n_A}(c^S_e, f^S_e) = \sqrt{\sum_{i=1}^{5} (c^S_{e,i} - f^S_{e,i})^2}, \tag{16}$$





where the superscript $S$ indicates a common scaling between zero and one. The aerosol composition $n_A$ for which the lowest euclidean distance is found, is used for any further steps of the ocean retrieval branch.

Once an aerosol composition has been found for each glint- and cloud-free pixel, OE is performed again. In this way, the pixel-wise AOT at 550 nm is estimated. In order to convert it to the AOT at 670 nm and 865 nm ($\tau_{A,\lambda}$), the pre-calculated normalized aerosol extinction coefficient $c_{n_A,\lambda}$ of the corresponding aerosol composition is used for the conversion, following:

$$\tau_{A,\lambda} = c_{n_A,\lambda} \cdot \tau_{A,550\,nm} \tag{17}$$

Finally, the Ångström parameter, which is strictly referring to the aerosol composition that has been used, is be estimated:

$$\alpha = ln\left(\frac{c_{n_A,\lambda_1}}{c_{n_A,\lambda_2}}\right) / ln\left(\frac{\lambda_2}{\lambda_1}\right) \tag{18}$$

### 2.3.4 Aerosol optical thickness retrieval over vegetated land surfaces

The availability of AOT over land is limited to dark vegetated surfaces, whereas bright surfaces will be excluded from any retrieval attempts. In particular, applicable land cover types are evergreen and deciduous broad and needle leaf as well as mixed forests, open and closed shrub lands, savannas, grassland, permanent wet and crop lands and mosaics of natural vegetation and crop land. These land surface types are based on the global land cover climatology by Broxton et al. (2014) that has been re-gridded to a spatial resolution of 30" (about 1 km). Very bright surface types (snow and ice, barren or sparsely vegetated) and water surfaces, as also present in this climatology, are not used in the retrieval. Hence, a pre-filtering of vegetated surface types is conducted before any OE land retrieval.

Additionally, the aerosol composition is assumed to be fixed following the MAC v1 aerosol climatology here. Even though, a good estimate of the aerosol component mixing ratios is as important over land as over ocean, the lack of a homogeneous surface over several land pixel, hinders attempts to use the same approach here.

The land forward operator has to account for the spectral surface reflectance in each MSI band in order properly describe the surface contribution to the TOA signal before any interpolation in the land LUTs can be done. This is achieved by using a surface parameterization (see sec. 2.3.5) in the M-AOT land forward operator in advance. The land state vector is not only composed of the AOT at 550 nm but includes the surface reflectance for MSI's VIS band and the shortwave normalized vegetation index ($NDVI_s$) in addition. Apriori estimates of these three elements are based on the MAC v1 climatology, black sky albedo of a 12 year (2002-2013) MODIS based climatology of bi-directional reflectance distribution function (BRDF) and albedo (Qu et al., 2022) and the ($NDVI_s$) estimation based on MSI measurements following:

$$NDVI_s = \frac{L_{c,SWIR-1} - L_{c,SWIR-2}}{L_{c,SWIR-1} + L_{c,SWIR-2}} \tag{19}$$

Consequently, the first state vector element is used to describe the atmospheric contribution and the latter two elements are used to describe the surface contribution to the TOA signal. In particular, the estimated surface reflectance for the VIS channel is used to determine the surface reflectance at other wavelengths taking variation of greenness into account by additionally



using the shortwave NDVI. Finally, the retrieved AOT at 550 nm is converted to AOT at 670 nm in the same manner as over ocean using Eq. 17.

### 2.3.5  M-AOT land surface parameterization

The description or separation of the surface and atmospheric, i.e. aerosol, contribution is the most challenging part in any imager AOT retrieval over land. It is more complex then over ocean because of the strong reflectance of the land surface in the
near- and short-wave infrared and due to the variability of surface reflectance for different surface types. Past studies for the MODIS aerosol product have shown already that an error of about 0.01 in the surface reflectance can lead to an error in the order of 0.1 in the retrieved AOT (Levy et al., 2007a). Hence, the surface reflectance has to be determined with high accuracy for the visible to shortwave infrared channels. The most suitable approach found for M-AOT, which ensures usability within an operational environment, reasonable run-times and delivers the best results based on pre-launch testing with simulated test
scenes and MODIS Level-1 data, is composed of two steps:

1. relating surface reflectance in a SWIR band to surface reflectance at shorter wavelengths by usage of extrapolation coefficients based on a MODIS climatology

2. transferring surface reflectance at any wavelength of the given climatology to MSI specific central wavelengths

Pre-calculated extrapolation coefficients used in the first step rely on a MODIS MCD43GF (Schaaf et al., 2002) based
climatology of BRDF and albedo (Qu et al., 2022). Hence, only MODIS bands 1 (645 nm), 2 (859 nm), 6 (1640 nm) and 7 (2130 nm) can be related using these coefficients. Further, it is assumed that the spectral behaviour of surface reflectance can be equally explained by usage of the black sky albedo. The empirically found formula to calculate extrapolation coefficients $d_{\lambda,\theta_s,cov}$, $e_{\lambda,\theta_s,cov}$ and $f_{\lambda,\theta_s,cov}$, based on this climatology using a ordinary least square fit, is:

$$\rho_\lambda = \rho_{b7} \cdot exp(d_{\lambda,\theta_s,cov}^{(NDVI_S + e_{\lambda,\theta_s,cov})} + f_{\lambda,\theta_s,cov}), \qquad (20)$$

where $\rho_{b7}$ is the black sky albedo at MODIS band 7, $\lambda$ corresponds to MODIS central wavelength, called here: $b1 : 645\,nm$, $b2 : 859\,nm$, $b6 : 1640\,nm$ and $NDVI_S = (\rho_{b6} - \rho_{b7})/(\rho_{b6} + \rho_{b7})$ is the NDVI in the shortwave infrared closest to MSI central wavelength, i.e. calculated based on MODIS band 6 and 7 black sky albedo. Since the black sky albedo itself is dependent on sun zenith angle $\theta_s$, this dependency has been preserved. Additionally, the best reconstruction of MODIS black sky albedo at shorter wavelength using band 7 was found when coefficients are dependent on the underlying surface cover type $cov$ (e.g. by
Broxton et al., 2014). In M-AOT, this formula and coefficients, which are stored in a LUT, are used as a first step in the surface parameterization scheme.

In the second step of the M-AOT surface parameterization, these surface reflectances for MODIS bands $\rho_{MOD}$ need to be transferred to surface reflactances at MSI central wavelengths $\rho_{MSI}$. Therefore, a linear model is applied using:

$$\rho_{MSI} = g_{MOD,MSI} \cdot \rho_{MOD} + h_{MOD,MSI} \qquad (21)$$



The underlying highly resolved surface spectra, which allowed for the calculation of $g_{\lambda_{MOD},\lambda_{MSI}}$ and $h_{\lambda_{MOD},\lambda_{MSI}}$, rely on the principal component approach that follows Vidot and Borbás (2014). They are saved in a LUT and are dependent on sun zenith angle and surface type as used for the empirical parameterization coefficients.

    The VIS surface reflectance, which is iteratively optimized during the OE steps, is technically transferred to the corresponding MODIS surface reflectance before the parameterization is applied, following:

$$\rho_{b1} = \frac{\rho_{VIS} - h_{b1,VIS}}{g_{b1,VIS}} \tag{22}$$

Afterwards, $\rho_{b1}$ can be applied in Eq. 20 to calculate surface reflectance at MODIS band 2, 6 and 7. These in turn have to be transferred to MSI central wavelength. In detail, the following linear model is used following Eq. 21:

$$\rho_{NIR} = g_{b2,NIR} \cdot \rho_{b2} + h_{b2,NIR} \tag{23}$$

$$\rho_{SWIR-1} = g_{b6,SWIR-1} \cdot \rho_{b6} + h_{b6,SWIR-1} \tag{24}$$

$$\rho_{SWIR-2} = g_{b7,SWIR-2} \cdot \rho_{b7} + h_{b7,SWIR-2} \tag{25}$$

    This implementation is subject to change in the future. In particular, the two step approach has the potential of being replaced by a simplified version, where only step one (Eq. 20) is applied. However, this would require a sufficiently big enough database of surface reflectances, specifically at MSI central wavelengths, first. This would allow to re-calculate extrapolation coefficients. One anticipated approach of building such a database could follow the proposed method in Jackson et al. (2013). Hence, it

could be build applying an atmospheric correction on MSI measurements over Aerosol Robotic Network (AERONET) sites. Alternatively, also the usage of ATLID aerosol products, once they have been validated post-launch, is imaginable to be used in such an atmospheric correction scheme.

## 3   Verification of M-AOT with extended simulated test scenes

Several nominal test scenes have been created for EarthCARE instruments (Donovan et al., 2022b; Qu et al., 2022) with
the intend of pre-launch verification of EarthCARE algorithms and algorithm chaining. In the scope of this study, the main advantage of the usage of these scenes is the possibility to verify the M-AOT algorithm performance and to evaluate estimated AOT with the fields used for the respective scene creation. These fields will be called truth or true fields or true AOT in this section. Estimated AOT of two of these test cases will be presented here in more detail as they include the largest number of cloud-free areas and contain land and ocean areas. The scenes always include aerosol and clouds and are named after their
location. They will be called Halifax and Halifax-Aerosol scene from now on.

    The presented AOT fields of the M-AOT product (Fig. 3a,c and 4a,c) rely on slight modifications of the operational algorithm regarding used aerosol LUTs and input meteorological fields, e.g. usage of wind speed as applied in L1 simulations instead of X-MET wind speed over ocean. This is done in order to use the same assumptions about aerosol optical properties and meteorological background conditions as have been used in the simulation of MSI L1c signals.

While test scenes are always processed as a whole with M-AOT, pixel-wise comparisons are only done for pixel for which the true AOT is at least double the true cloud optical thickness (COT) at 670 nm. Additionally, the maximum COT of each pixel's



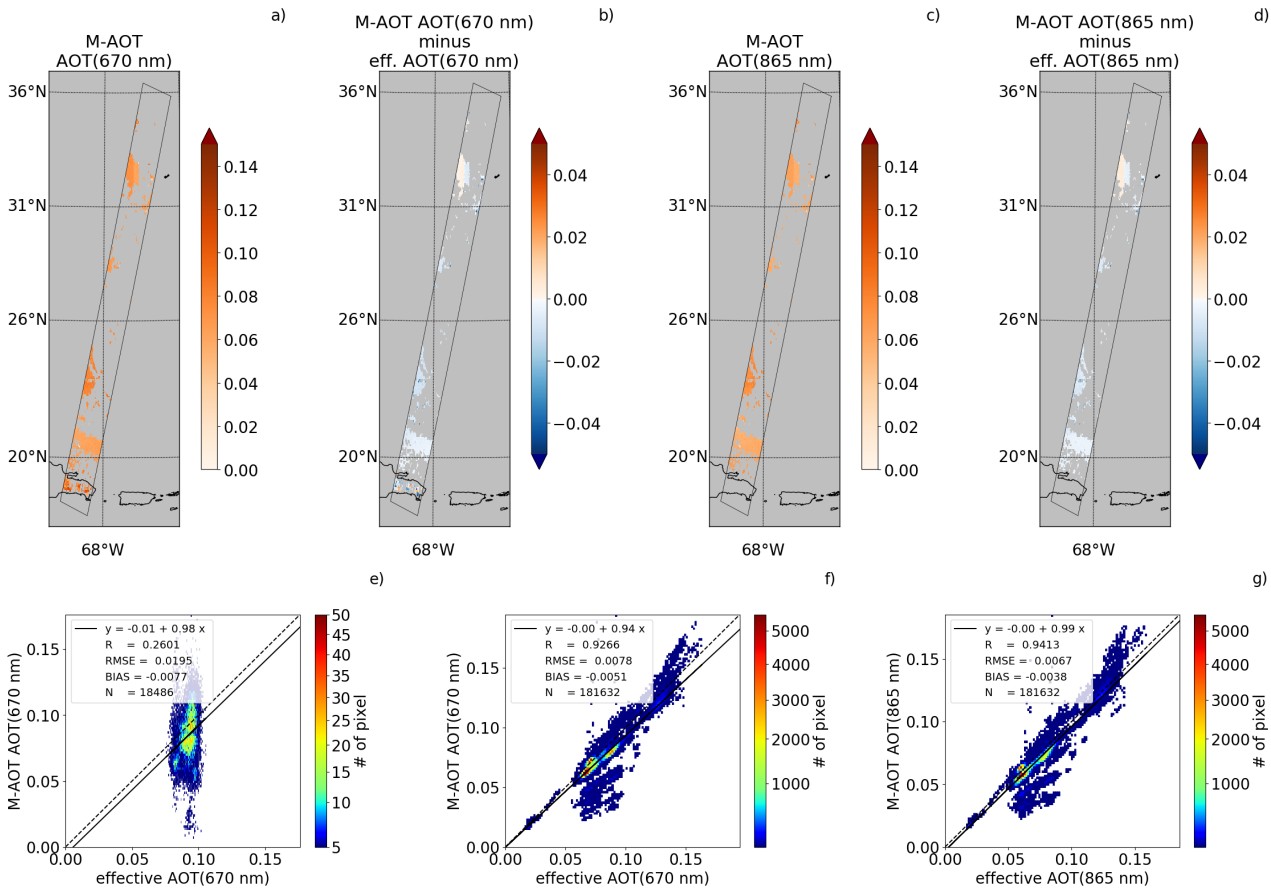

**Figure 3.** Retrieved AOT and comparison to effective AOT of true fields for the aerosol focussed part of the Halifax scene. Subfigures a) and c) show the M-AOT retrieved and successfully converged aerosol fields for all cloud-free pixel at 670 nm and 865 nm, respectively. Subfigures b) and d) show the differences between retrieved and effective AOT correspondingly. The lower panel subfigures show the pixelwise comparison between effective AOT (x-axis) and retrieved AOT (y-axis) at 670 nm over land (e), at 670 nm over ocean (f) and at 865 nm over ocean (g).



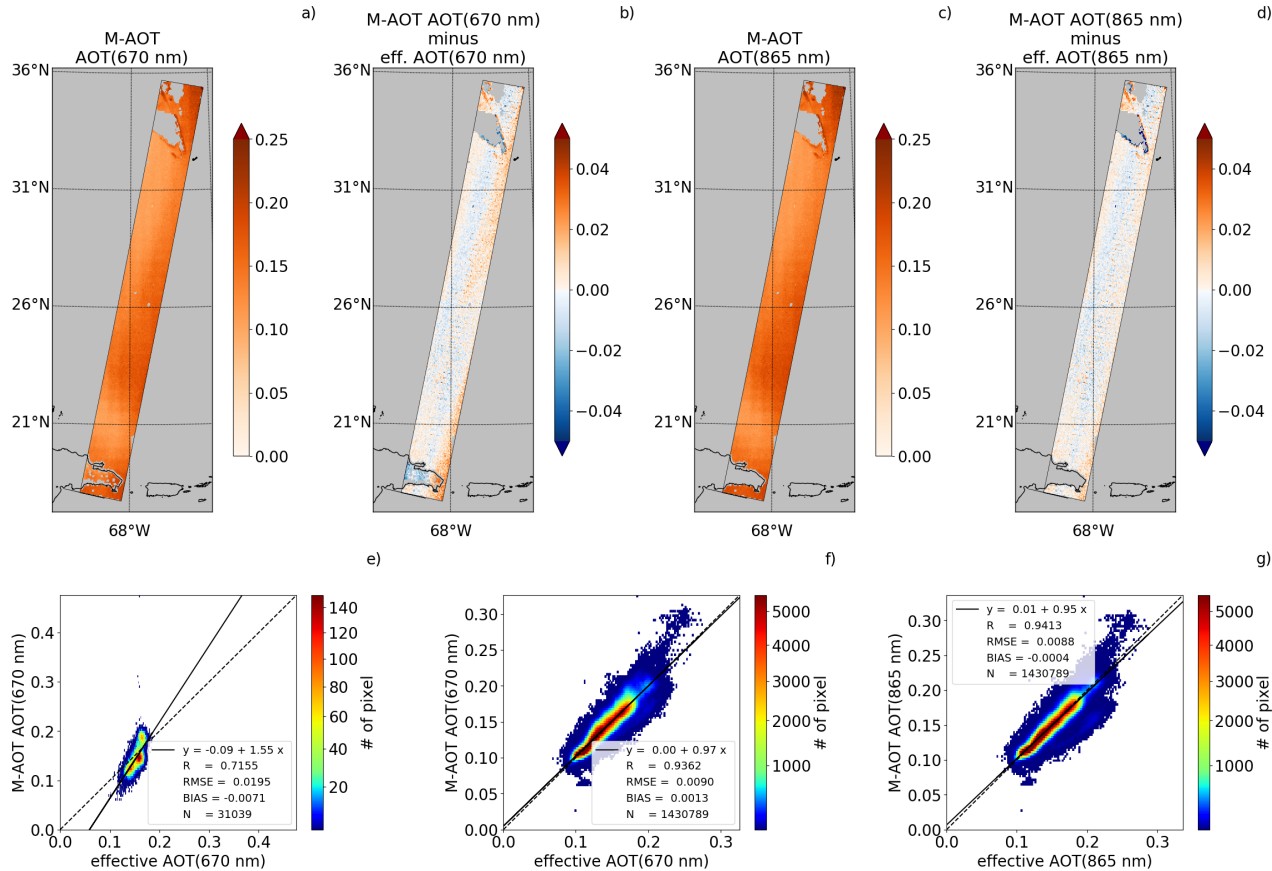

**Figure 4.** Same as Figure 3 but for the Halifax-Aerosol scene.

neighbours are checked for that criterion as well since actual instrument simulations were conducted on a higher resolved spatial grid before sampling them to MSI native grid. A potential mixing of residual cloudy and aerosol pixel can be reduced in this way. Further, the effective AOT is computed and used for pixel-based comparisons. It is defined here as the sum of true AOT and true residual COT, which might still be present even after filtering. Finally, potential scene artifacts in the test scene data caused by e.g. re-gridding around clouds or unrealistic sharp transitions in radiance are filtered. A pixel is excluded if its radiance at all four bands exceeds the median value plus or minus twice the standard deviation of the direct neighbours of that pixel. It is expected that at least 5 of the 9 potential pixels are available. After this kind of filtering, about 97.0% and 99.7% of pixel, for which M-AOT converged, are still used for further comparisons for the Halifax (Fig.3) and the Halifax-Aerosol (Fig. 4) test scene, respectively.

The Halifax scene starts over Greenland and ends South of the Dominican Republic. Since the North part of that scene is mostly cloud contaminated, only a smaller part of it is considered here. This scene is composed of mainly fine mode weakly absorbing and sea salt aerosol over glint-free ocean and parts of the Dominican Republic. Overall, the algorithm performance





lies within the goal mission requirement of an absolute accuracy of 0.02 for an integrated area of 10 x 10 km over ocean (Wehr,

2006) for this scene judging from the RMSE of 0.008 and 0.007 at 670 and 865 nm, respectively. The explained variance is

86% and 88% for the VIS and NIR bands. The agreement of M-AOT and effective AOT is worse over land surfaces than over

ocean. This is mainly caused by uncertainties in surface reflectance that lead to quite a large spread in resulting AOT.

The second case, presented here for algorithm verification, is the Halifax-Aerosol scene. It has not been created for a whole

EarthCARE frame but was specifically designed for aerosol focused testing of e.g. the M-AOT algorithm and subsequent

algorithms that use the M-AOT product, such as AM-COL (Haarig et al., 2022). The test data simulation is based on a step-

wise spectral behavior of surface reflectance, where the NIR to SWIR-2 bands all assume the same surface reflectance for

a pixel. This consequently lead to the need for an adaption of M-AOT to be run in a development mode in order to retrieve

AOT over land. The surface reflectance is prescribed in this mode since the surface parameterization used in M-AOT would

otherwise lead to a failed retrieval attempt for this kind of surface. The scene covers a 2000 km segment of the previously

presented nominal Halifax test scene. However, sea salt aerosol has been scaled by a factor of 2.5, while liquid clouds and

other aerosol types have been scaled by a factor of $10^{-6}$. This effectively leads to a scene consisting of ice clouds in the north

and sea salt aerosol everywhere else.

Overall, the retrieved aerosol loading at 670 and 865 nm agrees well with the true aerosol fields of the Halifax-Aerosol scene.

Slightly larger differences of the retrieved AOT at 670 nm as well as at 865 nm over ocean can mostly be found in the south-east

of the frame. The underlying issue is a wrong selection of the aerosol composition within the retrieval procedure (not shown).

The aerosol component mixing ratio to be best fitting to the measurements appears to be consisting of 85% salt and 15% fine

mode less absorbing aerosol in the east part over ocean leading to an overestimation of AOT. Nonetheless, the usefulness of

finding the best fitting aerosol mixture is still given for this scene since the climatological mixing would prescribe the mixture

to be mainly consisting of about 38% salt and 52% fine mode less absorbing aerosol. The overall performance appears to be

within EarthCARE requirements considering a correlation (Fig. 4 f-g) over water surfaces for that frame of 0.94 for AOT at

670 and 865 nm, respectively, and a RMSE of 0.009. Over land, the AOT retrieval performs better for the Halifax-Aerosol

scene than for the Halifax scene. This can be mainly explained by the correct knowledge of the underlying surface reflectance.

Differences in AOT are reaching up to -0.03 in the west part of the Dominican Republic and are slightly positive (0.02) in the

East under these assumptions.

350    Statistical measures of the presented scenes, as well as the overall comparison considering all scenes are summarized in Tab.

3. In general, the ocean retrieval branch delivers more accurate AOT estimates than the retrieval over land surfaces. Largest

differences reaching up to 0.02 or more are present if the used aerosol composition over ocean is not representative for the

actual mixing present in the scenes. Nonetheless, about 96% and 95% of retrieved AOT pixel are within the required accuracy

of 0.02 over ocean for these two presented test scenes.



**Table 3.** Comparison summary of statistical measures, i.e. linear regression coefficients, Pearson correlation coefficient (R), root-mean-squared-error (RMSE), bias, number of pixel used for comparision (N), and percentage of these pixel to have a difference in AOT of not more than +/- 0.02, for M-AOT applied to simulated test scenes

|  | case | linear regression | R | RMSE | BIAS | N | N within +/- 0.02 |
|---|---|---|---|---|---|---|---|
| AOT(670 nm) over water | Halifax (Fig. 3f) | $y = 0.00 + 0.94x$ | 0.93 | 0.008 | -0.005 | 181632 | 97.19% |
|  | Halifax-Aerosol (Fig. 4f) | $y = 0.00 + 0.97x$ | 0.94 | 0.009 | 0.001 | 1430789 | 96.28% |
|  | all simulated scenes (Fig. A3b) | $y = 0.00 + 1.00x$ | 0.98 | 0.009 | 0.001 | 2479055 | 96.36% |
| AOT(865 nm) over water | Halifax (Fig. 3g) | $y = 0.00 + 0.99x$ | 0.94 | 0.007 | -0.004 | 181632 | 97.56% |
|  | Halifax-Aerosol (Fig. 4g) | $y = 0.01 + 0.95x$ | 0.94 | 0.009 | -0.000 | 1430789 | 97.38% |
|  | all simulated scenes (Fig. A3c) | $y = 0.01 + 0.97x$ | 0.98 | 0.010 | 0.002 | 2479055 | 95.42% |
| AOT(670 nm) over land | Halifax (Fig. 3e) | $y = -0.01 + 0.98x$ | 0.26 | 0.019 | -0.008 | 18486 | 73.32% |
|  | Halifax-Aerosol (Fig. 4e) | $y = -0.09 + 1.55x$ | 0.72 | 0.019 | -0.007 | 31039 | 65.41% |
|  | all simulated scenes (Fig. A3a) | $y = 0.06 + 0.50x$ | 0.62 | 0.043 | 0.008 | 66988 | 46.53% |



## 4 Verification of M-AOT with MODIS test scenes

The specifics of the MSI instrument allow for a pre-launch testing of the algorithm with real-world data in addition to testing with simulated EarthCARE data. MODIS Aqua Level-1 data of collection 6.1 (**?**) has been chosen for that exercise due to the orbit specifics of an afternoon equator crossing time and channel settings close to the ones available from MSI. Radiance of band 1, 2, 6 and 7 are taken in this study in order to replace MSI signals for M-AOT testing. The cloud mask is taken from the corresponding official MODIS aerosol product (Levy et al., 2013, 2015). Meteorological input fields for M-AOT are replaced by ERA-5 reanalysis fields (Hersbach et al., 2020). Additionally, gas transmission LUTs have been modified to follow the spectral response of the respective MODIS bands instead of MSI filter functions. Similarly, aerosol LUTs as described in sec. 2.3.1 are replaced by LUTs specifically calculated for MODIS central wavelength. Finally, only the one step approach is used in the land forward operator surface parameterization since there is no need for a central wavelength correction of surface reflectance.

In total 2194 MODIS scenes from 2004 to 2021 (Fig. 5) have been processed with M-AOT in order to collect a statistically significant data base for the verification of the algorithm. The choice of scenes (Fig. 5 a) and c)) was focused on sampling the most common range of aerosol loadings over different seasons, increasing the number of potential match-ups with ship-borne based sun photometer measurements for ocean algorithm verification and using scenes for which a dark target like algorithm is suitable. However, even if provided with such an input database, only a subset will be usable within M-AOT due to e.g. cloud contamination, sunglint, snow, ice or too bright surfaces (i.e. deserts) still present in part of individual scenes or unsuccessful retrieval attempts. Hence, a fewer number of pixel will be ultimately available to be used further (Fig. 5 b)). Due to the usage of MODIS Level-1 scenes, the M-AOT product can be directly compared to the official MODIS product on a $10\,\text{km}$ x $10\,\text{km}$ grid in order to check for reasonable retrievals with M-AOT. The official MODIS Level-2 aerosol product is taken as reference here, since it has been widely validated in the past (e.g. Remer et al., 2005; Levy et al., 2010, 2013; Wei et al., 2019) and is considered to deliver more sophisticated AOT estimates, e.g. due to the availability of radiance measurements in more than the mentioned four bands used in the M-AOT algorithm.

For comparisons of individual scenes, as shown over the Iberian Peninsula (see Fig. 6), only M-AOT pixel are used for which the AOT retrieval has been flagged as successfully converged. MODIS AOT is only used within the comparison when the MODIS quality flag indicates a pixel as "good" or "very good".

Figure 6 shows the AOT at $670\,\text{nm}$ of M-AOT (a) and the official $10\,\text{km}$ MODIS AOT transferred to $670\,\text{nm}$ (b) in the upper panel and at $865\,\text{nm}$ in the lower panel that spans a wider range of aerosol loading than previously demonstrated with simulated test scenes. The AOT of the MODIS aerosol product has been transferred via the Ångström parameter, which is based on the official MODIS AOT at $550\,\text{nm}$ and $660\,\text{nm}$ over land and at $660\,\text{nm}$ and $860\,\text{nm}$ over ocean. Both fields of aerosol loading show a similar pattern over land and ocean. However, the M-AOT product does contain AOT for less pixel over land. The direct comparison of both products (see Fig. 7) has been done by using the median of M-AOT AOT for the $10$ x $10$ pixel around a MODIS product pixel. The correlation for that scene is 0.97 and 0.99 over ocean and still reaches 0.87 over



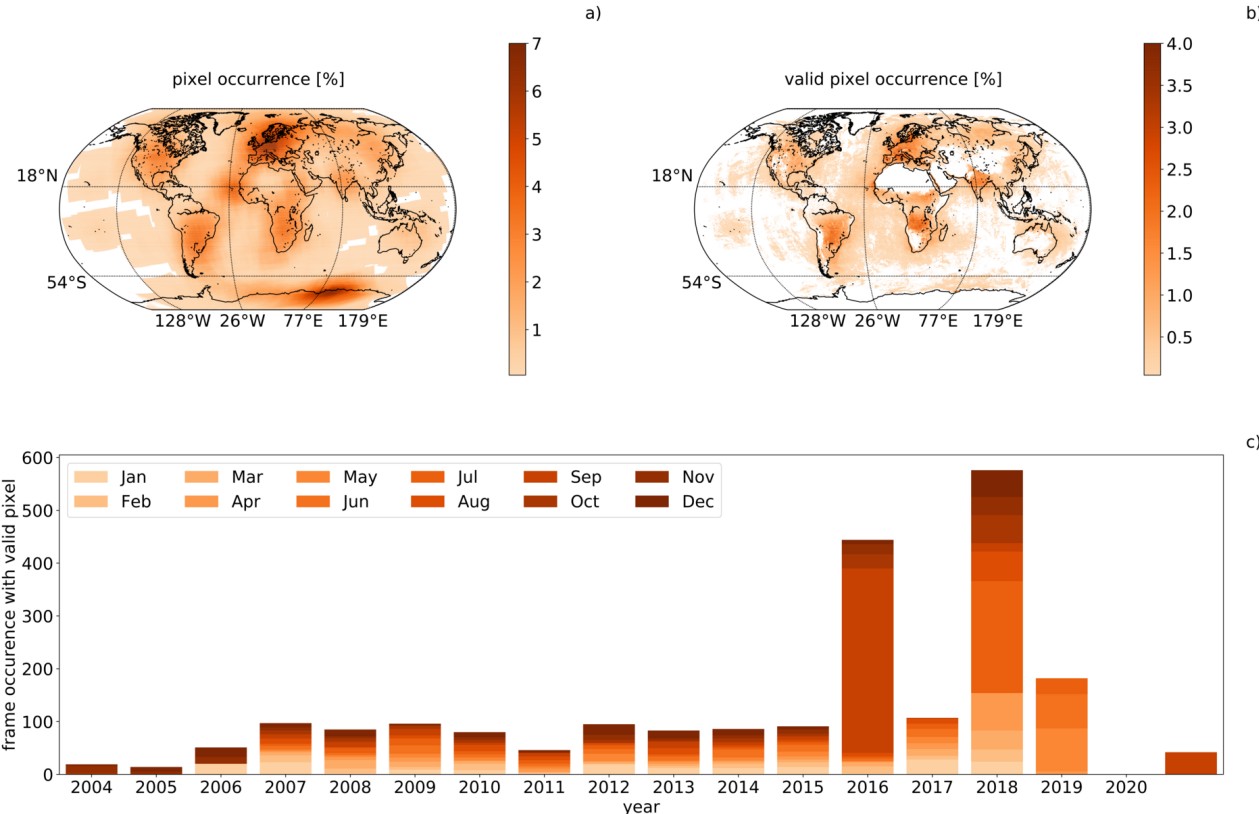

**Figure 5.** Scene occurence of used MODIS test data. Subfigure a) shows the overall occurence, b) the used pixel occurence in percent. Subfigure c) shows the total number of frames used per year (x-axis) and month (colored).

land, even though AOT is mostly overestimated by M-AOT there. The RMSE of 0.02 over ocean at 865 nm again fulfills the AOT accuracy requirement. The RMSE of AOT at 670 nm is slightly higher with 0.04.

While the inter-comparison of one scene can hint at specific local shortcomings, a statistical comparison of all scenes gives a more complete picture of the M-AOT algorithm performance. Therefore, all valid M-AOT AOT match-ups have been compared with all "good" or "very good" MODIS AOT for all processed scenes and are shown in Fig. 8.

Once again, the agreement of M-AOT and MODIS is better over ocean than over land surfaces and AOT at 865 nm has a slightly higher agreement than at 670 nm over ocean considering an RMSE of 0.10 (land), 0.029 (AOT(865 nm), ocean) and
0.039 (AOT(670 nm), ocean). Deviations are in particular increasing for higher AOT (>2) over ocean and can be seen in the two branches over- and underestimating AOT. This hints at different assumptions in the respective retrieval algorithms, such as about the underlying aerosol type.

Finally, M-AOT AOT is compared to AERONET v3 (Giles et al., 2019) over land and Maritime Aerosol Network (MAN, Smirnov et al., 2009) data over ocean in order to quanitify the overall performance. Since sun photometer based measurements
represent point-like measurements with a higher temporal resolution compared to satellite based products, M-AOT pixel are





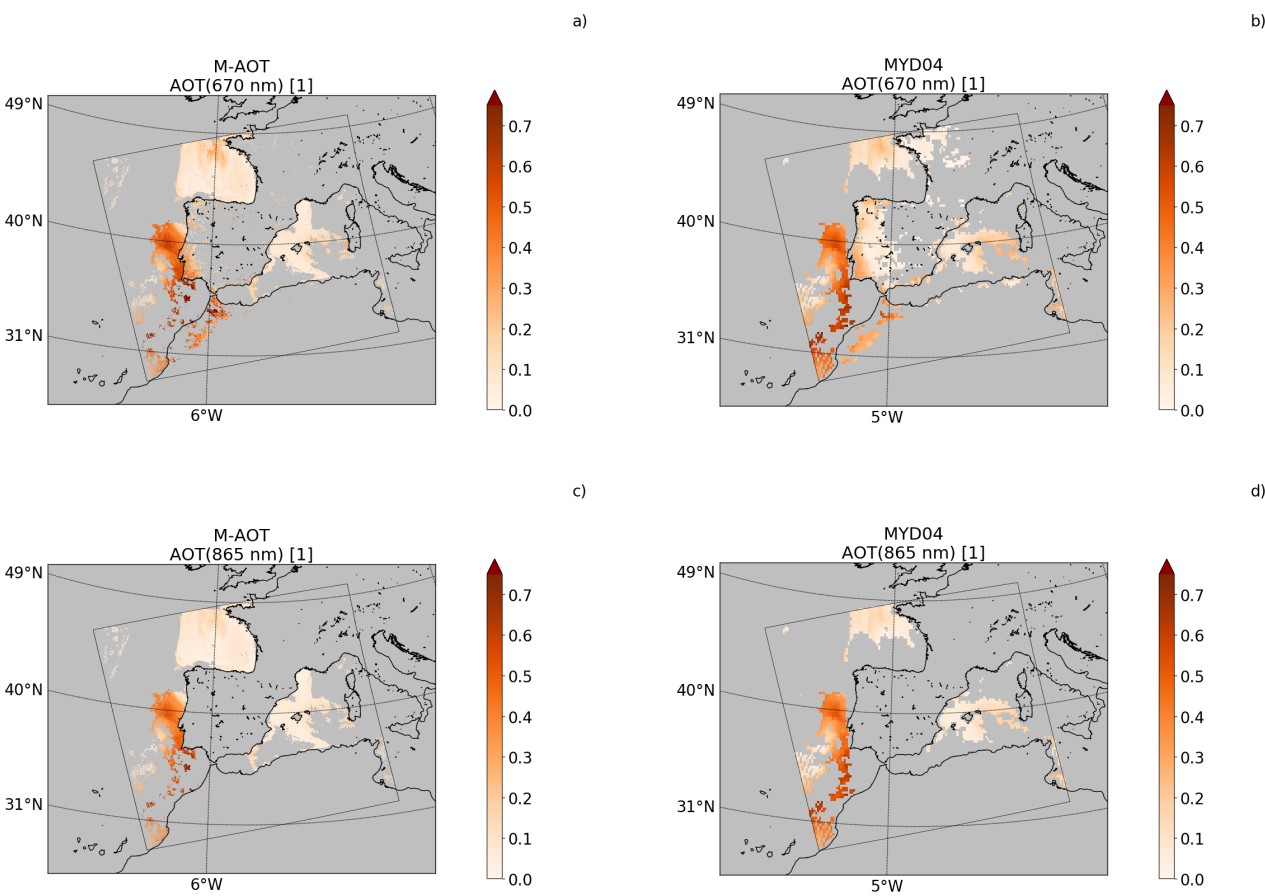

**Figure 6.** AOT at 670 nm from M-AOT (a) and based on the 10 x 10 km MODIS MYD04 product (b) and correspondingly at 865 nm from M-AOT (c) and MODIS (d) over the Iberian Peninsula at 20 March 2009 at 13:20 UTC.

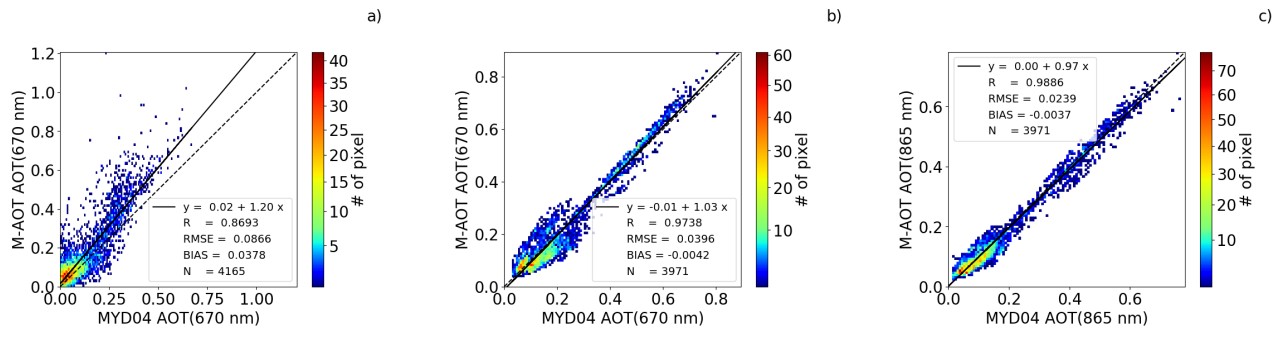

**Figure 7.** Comparisons of M-AOT (y-axis) and MYD04 (x-axis) product for the scene shown in Fig. 6 of AOT at 670 nm over land (a) and ocean (b) as well as at 865 nm over ocean (c).





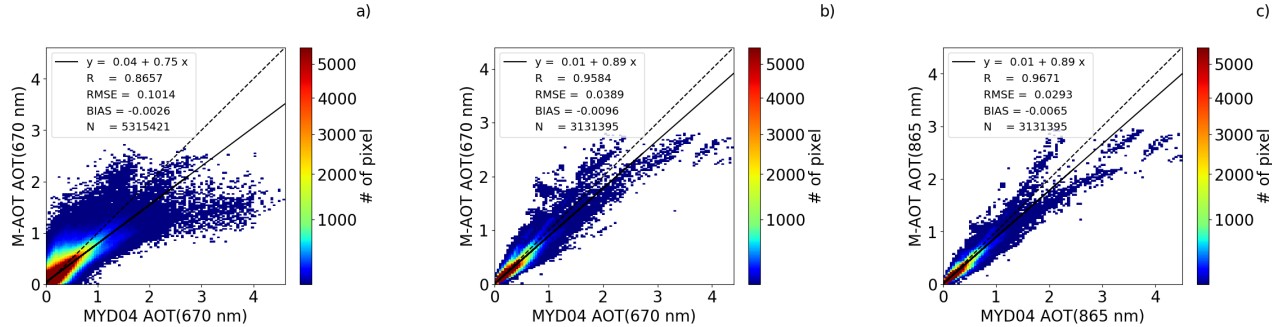

**Figure 8.** Comparisons of M-AOT (y-axis) and MYD04 (x-axis) product for all processed scenes with M-AOT of AOT at 670 nm over land (a), at 670 nm over ocean (b) and at 865 nm over ocean (c). Regression curves and statistical measures are based on outlier cleared matches, meaning all values that are outside ±3 standard deviations of the differences between MODIS and M-AOT AOTs.

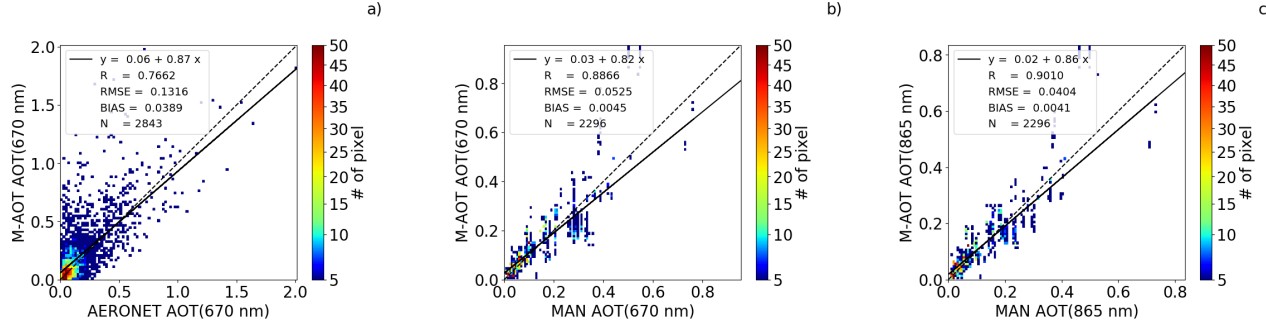

**Figure 9.** Comparison of M-AOT (y-axis) and AERONET aerosol optical thickness at 670 nm over land (a), Maritime Aerosol Network (MAN) aerosol optical thickness at 670 nm (b) and 865 nm (c). Regression curves and statistical measures are based on outlier cleared matches, meaning all values that are outside ±3 standard deviations of the differences between MODIS and M-AOT AOTs.

only considered for comparisons if the MODIS measurement was taken within 30 minutes and in a search radius of 0.045° (5 km) around an AERONET or MAN measurement. M-AOT AOTs have been accumulated and the median is taken for further comparisons. Additionally, corresponding in-situ AOTs have been transferred from wavelength reported in AERONET and MAN to 670 and 865 nm using the Ångström parameter.

Overall, the verification measures with AERONET and MAN (Fig. 9) are of a similar magnitude as for the MODIS comparison even though slightly inferior to them. Statistical measures are summarized for MODIS Level-1 input based comparisons in Tab. 4. The worse result of AOT(670 nm) compared to AOT(865 nm) might be a consequence of neglecting any water leaving reflectance in the visible band (MODIS band 1). While this was acceptable for EarthCARE simulated based test scenes, it becomes more important for real-world data. Hence, when it is known that underlying chlorophyll concentrations are enhanced,

M-AOT AOTs should be used with caution since it could, on the one hand, have consequences on the aerosol composition choice and, on the other hand, increase estimated AOT at 670 nm.



**Table 4.** Summary of statistical measures of M-AOT for MODIS MYD04, AERONET and MAN comparison

|  | case | linear regression | R | RMSE | BIAS | N | N within +/- 0.02 |
|---|---|---|---|---|---|---|---|
| AOT(670 nm) over water | MYD04 example (Fig. 7b) | $y = -0.01 + 1.03x$ | 0.97 | 0.040 | -0.004 | 3971 | 41.78% |
|  | MYD04 all scenes (Fig. 8b) | $y = 0.01 + 0.89x$ | 0.96 | 0.039 | -0.010 | 3069744 | 54.65% |
|  | MAN (Fig. 9b) | $y = 0.03 + 0.82x$ | 0.89 | 0.052 | 0.004 | 2296 | 43.68% |
| AOT(865 nm) over water | MYD04 example (Fig. 7c) | $y = 0.00 + 0.97x$ | 0.99 | 0.020 | -0.004 | 3971 | 64.57% |
|  | MYD04 all scenes (Fig. 8c) | $y = 0.01 + 0.89x$ | 0.97 | 0.029 | -0.006 | 3065174 | 66.91% |
|  | MAN (Fig. 9c) | $y = 0.02 + 0.86x$ | 0.90 | 0.040 | 0.004 | 2296 | 50.57% |
| AOT(670 nm) over land | MYD04 example (Fig. 7a) | $y = 0.02 + 1.20x$ | 0.87 | 0.090 | 0.038 | 4165 | 29.03% |
|  | MYD04 all scenes (Fig. 8a) | $y = 0.04 + 0.75x$ | 0.87 | 0.101 | -0.003 | 5262684 | 21.49% |
|  | AERONET (Fig. 9a) | $y = 0.06 + 0.87x$ | 0.77 | 0.132 | 0.039 | 2843 | 21.03% |

## 5 Discussion

In this study, the algorithm behind the EarthCARE Level-2 imager aerosol product M-AOT is presented with the intention of highlighting assumptions made and limitations present. The product consist of AOT at 670 nm and 865 nm over ocean and of AOT at 670 nm over land on native MSI spatial resolution. As common for imager based aerosol products, AOT is only available for cloud-free, day-time conditions and if the separation of surface and atmospheric contributions is possible to a reasonable degree. M-AOT has been applied to simulated EarthCARE MSI Level-1 and MODIS Level-1 data. First verification studies have shown that the product's AOT is acceptable given the limited information available from MSI. Based on verification tests using simulated MSI inputs, an accuracy of 0.02 could be reached over ocean. This will have to be confirmed during the commissioning phase when real world EarthCARE MSI data will be available. The product should be used with caution close to cloud edges, in presence of high chlorophyll over water and close to coastal areas. Over land surfaces, pixel for which the retrieved surface reflectance is relatively bright at 670 nm are flagged. This flag is part of the products quality status, which enables users to filter out such pixel.

It is likely that several configurable parameters, which have been proven reasonable for pre-launch studies, will be adjusted during the commissioning phase. Further, the M-AOT processor offers the option to exchange several of the many auxiliary data. In particular, several of the made assumptions can be modified and optimized in order to account for findings during the commissioning phase to improve product accuracy. For example, the surface parameterization coefficients are very likely in need for an update once real world data is available. This could be done by replacing the underlying climatology with another one, e.g. by building an MSI based climatology in the future. Similarly, a priori knowledge about aerosol optical thickness or the aerosol component mixing could be replaced by a more recent climatology or by reanalysis or forecast fields, for example



from Copernicus Atmosphere Monitoring Service (CAMS, Inness et al., 2019). This setting might not be applicable in an operational setting but might lead to improved results in an offline reprocessing.

Additionally, the spectral wavelength shift in dependence of viewing zenith angle within MSI bands (Wehr et al., 2022) are under investigation regarding the impact on the M-AOT product. The aim of these studies is to find a mitigation approach in the M-AOT algorithm itself that will account for it. At this moment, it is thought of expanding current auxiliary input data, e.g. aerosol LUTs, gas correction coefficients and surface parameterization correction coefficients to directly account for varying wavelength. While the approach is not part of this study and can only be published at a later point in time, it should be kept in mind that this could be a potential error source for MSI based AOT products, if it is not accounted for in the algorithm in the future. In particular, over land surfaces, this will become of importance since already small errors in surface reflectance assumptions have an influence on the retrieved aerosol loading.

*Code and data availability.* The M-AOT processor used in this study is intended to be made available after the commissioning phase of EarthCARE. All datasets from various instruments and simulated scenes are publicly available. The EarthCARE Level-2 demonstration products from simulated scenes, including the M-RGR, M-CM and M-AOT product discussed in this paper, are available from https://doi.org/10.5281/zenodo.7117115 (van Zadelhoff et al., 2022). MODIS Level-1 (MODIS Science Data Support Team (SDST), 2017) and the MODIS Level-2 aerosol product MYD04 (Levy and Hsu, 2015) are available through the Level-1 and Atmosphere Archive  Distribution System Distributed Active Archive Center (LAADS DAAC: https://ladsweb.modaps.eosdis.nasa.gov, last access 31-01-2023). ERA-5 reanalysis data is available from the Copernicus climate data store (https://cds.climate.copernicus.eu/cdsapp#!/home, last access 31-01-2023). AERONET and MAN data are available through the AERONET website (https://aeronet.gsfc.nasa.gov/new_web/aerosols.html, last access 31-01-2023) and AERONET MAN website (https://aeronet.gsfc.nasa.gov/new_web/maritime_aerosol_network.html, last access 31-01-2023), respectively.

## Appendix A:  Verification of M-AOT using further EarthCARE simulated test scenes

The following examples show M-AOT AOT (Fig. A1, A2) based on the two remaining simulated test scenes that have been produced for verification studies. They are, again, called according to their location: Baja and Hawaii scene. For completeness, simple differences between effective AOT and a comparison summarizing all scenes (Fig. A3).

*Author contributions.* ND, RP, FF, JF conceptualized and drafted the methodology of the MSI aerosol algorithm. ND, RP, implemented and processed the various data sets. ND, RP performed the verification studies based on simulated EarthCARE and MODIS test scenes. ND, RP, LK created the draft manuscript. LK created the transmission / gas correction LUTs. ND, FF carried out radiative transfer simulations, defined and built land and ocean LUTs. FS did the pre-processing of MODIS Level-1 test scenes to be ingestible by the M-AOT processor. JF supervised this study. All authors were involved in discussions during the M-AOT development and contributed material and/or text to the manuscript.

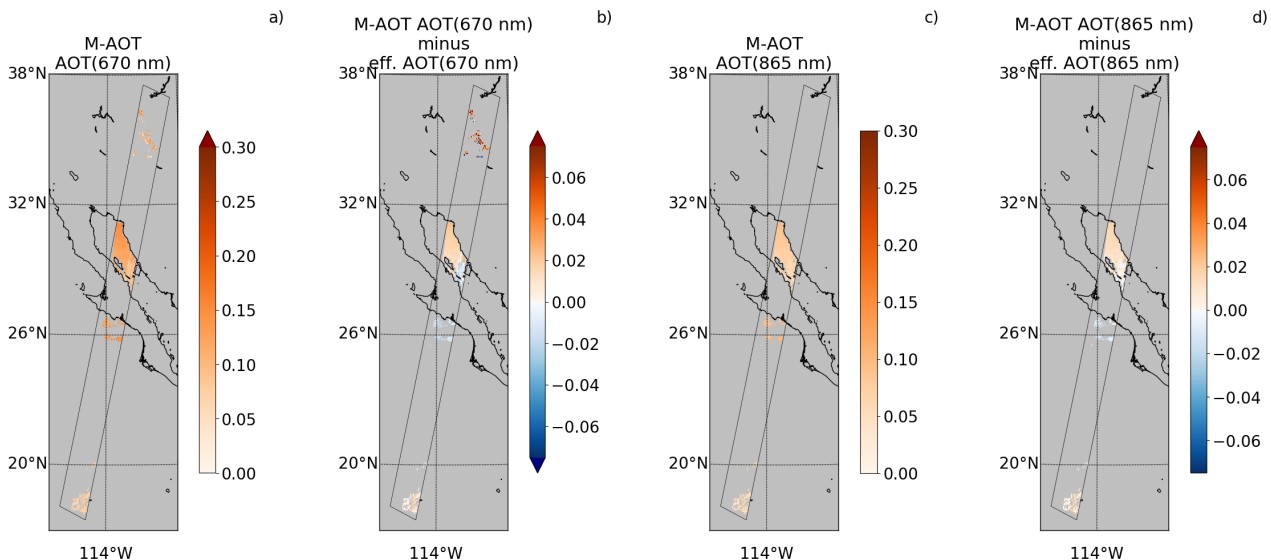

**Figure A1.** Retrieved AOT and comparison to effective AOT of true fields for the aerosol focussed part of the Baja scene. Subfigures a) and c) show the M-AOT retrieved and successfully converged aerosol fields for all cloud-free pixel at 670 nm and 865 nm, respectively. Subfigures b) and d) show the differences between retrieved and effective AOT correspondingly.

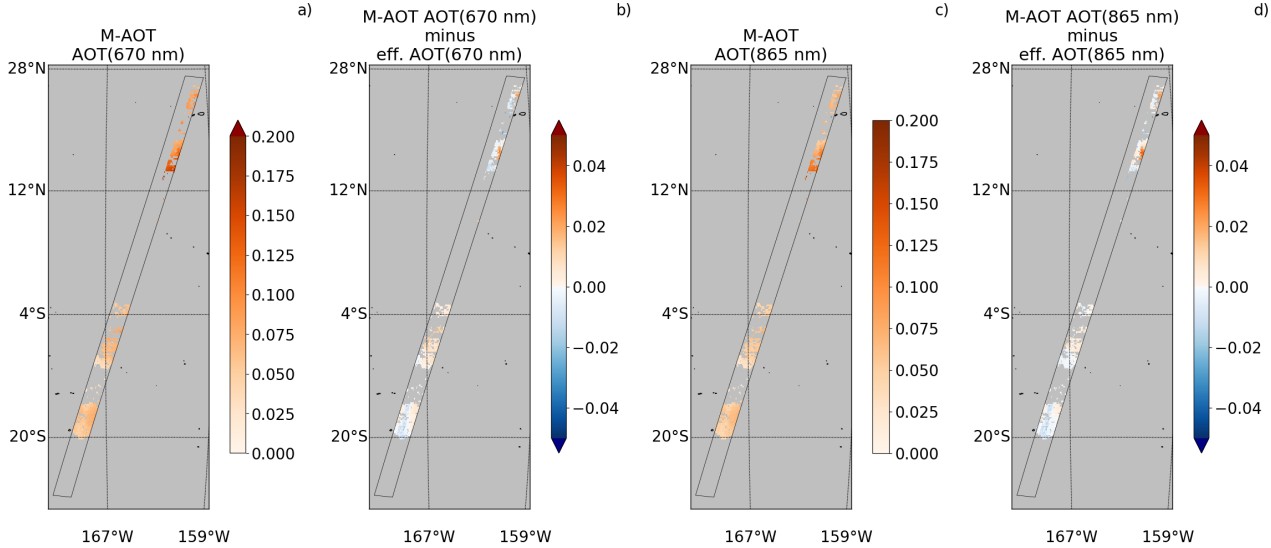

**Figure A2.** Same as Fig. A1 but for the Hawaii scene.



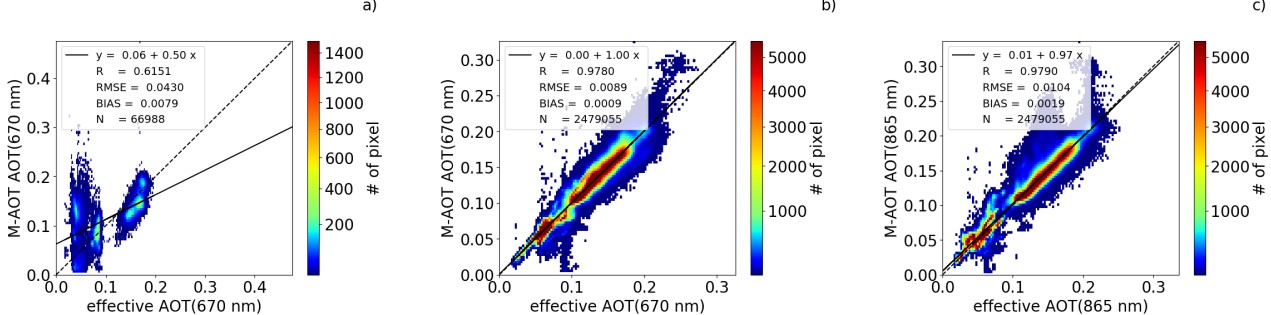

**Figure A3.** Pixel-wise comparison between effective AOT (x-axis) and retrieved AOT (y-axis) at 670 nm over land (a), at 670 nm over ocean (b) and at 865 nm over ocean (c) for all simulated EarthCARE test scenes (i.e. Halifax, Halifax-Aerosol, Baja and Hawaii).

*Competing interests.* The authors declare that they have no competing interests.

*Acknowledgements.* This work has been funded by ESA grants 4000112018/14/NL/CT (APRIL) and 4000134661/21/NL/AD (CARDI-NAL). The authors thank Tobias Wehr and Michael Eisinger for their support over many years and the EarthCARE developer teams for valuable discussions in various meetings. The authors would like to express their gratitude to the MODIS Science Team, the AERONET and
AERONET/MAN networks for providing their data to the scientific community. We thank the respective PIs and members of the teams for establishing and maintaining the respective instruments and data products over many years.



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
