# Peer review of "Aerosol optical depth retrieval from the EarthCARE multi-spectral imager: the M-AOT product"

_EGUsphere, 2023_

## Author Comment (AC1)

**Reply to Anonymous Referee #1**

We thank the reviewer for their helpful and constructive comments. Please find our answers (black text) to the comments (blue text) in-line below. Respective changes are indicated in the revised manuscript in blue and are stated here in addition when a reviewer comment lead to a modification of the manuscript along with the updated line number, where necessary.

This paper describes the aerosol optical depth retrieval candidate algorithm for the EarthCARE mission. The method described in the paper shows quite some limitations, mostly on the limited number considered valid for retrieval purposes. The results presented should be further discussed and the algorithm should be better placed in a broader context, highlighting its unique features, pros and cons, compared to other existing algorithms. Overall, the paper could be strongly improved with more precise explanation and much more elaborated discussions.

Thank you for your comment.

While we fully understand that an elaborated assessment of algorithm pros, cons and unique features are typically highlighted and compared in publications of novel algorithms, this publication is intended to serve as an introduction to the operational M-AOT product. Hence, the algorithm is introduced as a means to understand limitations this product has, not as a novel algorithm itself.

We rephrased the following statement in the beginning of the introduction for clarification and in order to avoid confusion, we might have introduced before:
"The presented algorithm and its accompanied product provide conventional imager aerosol information within the retrieval chain (Eisinger et al., 2022)"
to →
"The operational imager aerosol product, that is introduced here, provides aerosol information based on passive measurements alone within the Level 2 retrieval chain (Eisinger et al., 2022), using a conventional passive aerosol retrieval algorithm approach. Both, product and algorithm, are [...]" (L25-28)

Further we added a dependent clause in the last paragraph of the introduction:
"The operational Level 2 M-AOT algorithm […] is introduced [...] in order to highlight the aerosol product characteristics [...]" (L71)

Additionally, we think a point we missed to make right in the beginning of our manuscript are the constraints the operational M-AOT algorithm and consequently also the official Level-2 product, also called M-AOT, are exposed to due to the nature of the instrument and the operational requirements of the ESA ground segment. Hence, we added the following statement in the introduction for clarification:
"While M-AOT has been developed to operationally enable users interested in aerosol properties from EarthCARE to assess the horizontal aerosol loading, it is subject to many constraints due to the design of MSI itself, e.g. number and placement of spectral channels, small swath width, no polarization or multi-angle capabilities, as well as due to operational, near-real-time algorithm requirements imposed by the ground segment. Both, instrument design and computational constraints, prevent more elaborate retrieval attempts using for instance multi-temporal approaches, as used for e.g. SEVIRI and PROBA-V (Luffarelli and Govaerts, 2019). The usage of real-time a priori updates of made assumptions in the

M-AOT algorithm, e.g. about land surface characteristics or the aerosol composition, is not possible due to operational environment definitions. Further, in order to provide the product in near-real-time along with other Level 2 products, there are strict run-time requirements and processing hardware assignments in place." (L61-69)

We additionally added the purpose of MSI in the discussion section of the manuscript once again to remind the reader of what was already stated in the introduction:
"Nonetheless, it could be demonstrated that the M-AOT product is capable of serving its purpose of offering a horizontal context by providing columnar aerosol loading information." (L462-463) after stating the strict limitations (L458-461).

We hope that these modifications are sufficient for the purpose of a product introduction publication now. Further, we think that statements about MSI, using e.g. "conventional", "heritage" or "limited" instrument, should be enough information for a the reader or potential user of the product. Further, we think that placing M-AOT in a broader context of passive imager retrievals, highlighting pros and cons could potentially end up in a review paper on passive aerosol retrievals due to MSI's limitations. However, this is clearly not the scope of this publication.

General comments

- The verification of a retrieval algorithm against simulated data is expected to represent the best possible retrieval performances, as all the assumptions are the same both in forward and inverse modelling and as the true state of the scene is known, which is never the case from actual observations. Therefore I am a bit puzzled to see such low correlation against the simulated test data set. Either there is something wrong in the construction of this exercise, or there is something I am missing.
  While we fully agree, that the verification of the retrieval algorithm using simulated test data should represent the best possible retrieval performance, it should be kept in mind here that the simulated test data and the M-AOT algorithm do not use the same forward models. As this was not clearly stated before, we added an additional statement for clarification in L332-336:
  "Even though, input assumptions are tried to be used as consistently as possible, it should be kept in mind that the L1c forward model is not the same as the M-AOT forward model. Hence, no perfect agreement should be expected for this kind of verification. In particular, the surface spectrum over land surfaces is quite different in the respective forward models. While M-AOT uses the parameterization described in sec. 2.3.5, the surface description used for the test scene creation is based on Vidot and Borbás (2014)."
  We omit a full list of differences here, since this would be out of scope of this work. Nonetheless, we agree that the addition of the cause for the biggest discrepancies over land surfaces, should be added and hence did so.

- The selection of valid pixels is mentioned in several part of the manuscript, yet it remains confusing. To summarise, the algorithm only processes open ocean water and dark vegetation, is this correct? If so, it should made it clear in the paper. Sometimes you refer to "relatively bright" or similar statement that are not very conclusive.

You are correct with your assumption: only ocean and dark vegetation. We removed "relatively" and "very" bright or dark wherever it was unnecessarily used.

*Also, in Table 1 the values you chose for the surface reflectance in the NIR are quite low, I assume this is because dry vegetation is also excluded from the processing. If this is not the case, you should include larger value of surface reflectance.*
Again, you are correct with your assumption. Dry vegetation is excluded. We modified the first sentence in sec. 2.3.4 "Aerosol optical thickness retrieval over vegetated land surfaces" to explicitly exclude dry vegetation. (see L245).
Additionally, we are sorry to have introduced confusion by accidentally having switched the superscript descriptions of 1 and 2 in Table 1, which is corrected now. In particular: 1: for VIS, SWIR-1 and SWIR-2, 2: for NIR only

*Also, in what quantity is the latter expressed? BRF, BHR, ecc.. Please clarify.*
When we state surface reflectance, we are considering a Lambertian reflector. This is already stated in sec. 2.3.1 Forward model. In addition, we further clarify the usage of bi-hemispherical reflectance in sec. 2.3.4 now. This update is also related to a minor comment or yours below.

- *State vector: it is not clear what does the algorithm actually retrieve: the AOT at 550 nm (as stated at L163) or the AOT at 670 nm and (over water) at 865 nm, as suggested from the abstract and introduction? I seem to understand that the retrieved quantity is the AOT at 550nm, it should therefore be made clear that the AOT at 670 and 865 nm is derived from 550 nm based on your (quite strict) aerosol composition assumptions).*
  We added:
  "Hence, AOT will need to be extrapolated to 670 nm and 865 nm using strict assumptions about the underlying aerosol composition." (L177)

- *I think that there is a bit of confusion between absolute accuracy and RMSE, which is dependent on the magnitude of the quantity you're measuring. The RMSE can largely increase with the AOT range, it should not be taken as a reference for absolute accuracy purposes.*
  We added a clarification and your warning in the updated manuscript:
  "It should be noted that the term absolute accuracy as used in the formal mission requirements inherits some ambiguity. Added on top, the RMSE, which is used as a measure for it here, includes both systematic and random errors and is dependent on the magnitude of AOT itself in the comparisons. Nonetheless, keeping this in mind, this terminology will be used from now on in this study as an initial and rough accuracy assessment before launch." (L355-359)
  Additionally, we also state other quality measures in the verification sections.

- *The impact of your assumptions should be better assessed, e.g. lambertian surfaces, fixed atmospheric composition, etc.*
  We decided to not do a full assessment of the assumptions since this would be out scope and decided to rather list them throughout the manuscript. However, an elaborated assessment of the respective assumptions should rather be done based on real world EarthCARE MSI data.

- L4 670 nm over ocean and valid land pixels, and at 865nm over ocean.

  Thank you, done.

- L36 It has been applied, for instance, to MODIS

  Thank you. Updated.

- L40 The algorithm from Luffarelli and Govaerts is not satellite dependent. In the very same paper you refer to it is applied to PROBA-V as well. Please correct.

  We added both applications of the Luffarelli and Govaerts algorithm explicitly in L65, where it fits better in the logical structure of our text.

- L52 Please update the reference to Govaerts and Luffarelli, 2018

  Updated. Now, Using the Govaerts and Luffarelli, 2018 instead of the Govaerts et al., 2010.

- L55 "where possible" is very vague. Please be more clear: "on valid land pixels", "on dark vegetated surfaces"

  Rephrased: "over dark vegetated land pixel". (L57-58)

- L79 Add reference to the DEM model used.

  Updated.

- L134 Explain why the method to describe water bodies is not suitable for coastal water. I assume it is because a fix value of chlorophyll content is considered?

  Added an additional sentence for clarification:

  "This means, only a clear water spectrum is assumed for LUT simulations since there is operationally no real-time information available about e.g. chlorophyll content or colored dissolved organic matter or sediment." (L146-148)

- L143 […] the AOT, the aerosol scattering phase function […]

  Thank you. Updated.

- L170 make sure you use the same symbols as in Eq. 8

  Thank you. Fixed.

- L171 Therefore, the state vector […]

  Updated.

- L175 justify the value of 0.03 and 0.001. Do they come from a certain number of tests, from literature, …?

  Added:

  "These two values are based on pre-launch testing of the algorithm. Nonetheless, they might be modified based on commissioning phase algorithm testing." (L188-191)

- L242 surface reflectance in terms of? Albedo?
  Yes. We further clarified by adding "[…] the surface reflectance, in terms of spectral bi-hemispherical reflectance or albedo, [...]" (L258), which is also related to one of your general comments above.

- L254 You should mention the issue of the radiative coupling between surface and atmosphere
  Added:
  "Hence, the TOA signal is not dominated by atmospheric scattering processes as over ocean outside of sun glint, but rather represents a strongly coupled surface-atmosphere signal." (L272-274)

- L254 more complex than
  Thank you. Adjusted.

- L267 "empirically found". How? What is the accuracy?
  We added a clarification. Reading:
  "This formula has been empirically chosen in such a way, that MODIS band 7 black sky albedo can be used to reasonable reproduce the expected black sky albedo at band 1, 2 and 6. The correlation between parameterized and actual black sky albedo is above 0.9 for band 1 and 6 and above 0.7 for band 2. The corresponding average root mean squared error is below 0.02 (band 1 and 6) and 0.04 (band 2)." (~L293-296)

- Eq. 21 make sure to define all symbols
  Updated.

- Figure 3: the scatterplots show values close to 0.15 which are not really visible in the images. Also there is a cloud of points in figure 3 f) and 3 g) where the M-AOT strongly underestimate AOT between 0.05 and 0.1. Do you have a possible explanation for this?
  Updated the figure to a higher resolution and added the following clarification:
  "The bulk of the aerosol loading is to be expected in the range 0.06-0.1 for the Halifax scene. This is also color-coded in the respective comparison figures e-g). Nonetheless, occasionally, values exceeding this range are also available here although scattered, mostly close to cloud edges and consequently hard to spot in subfigures a) and c)." (L350-L353)
  "Strong underestimations in AOT over ocean between 0.05 and 0.1 are present close to cloud edges according to Fig. 3b, d, f, g). This is caused by a wrong aerosol composition assumption following the approach of sec. 2.3.3 in such complex regions, where, contrary to the assumptions used, in reality, a sharp transition of optical properties occurs." (L361-364)

- L325 "Explained variance"??
  Added clarification:
  "The explained variance, in terms of the squared correlation in percent, is [...]" (L359)

- Table 3 Here you suddenly mention "all simulated scenes". You should mention in the text that more scenes were simulated and refer to the annex.

  Added sentence after introduction of the test scenes referring to the appendix.

  "Nonetheless, results for all simulated test scene cases will be summarized at the end of this subsection and corresponding figures can be found in appendix A." (L326-327)

- L357 Missing reference

  Thank you. Updated.

- L376 "more sophisticated" More than what?

  Updated. "than M-AOT applied to MODIS" (L414)

- L381 transferred = extrapolated?

  Exactly, updated.

- L387 The correlation for that scene is 0.97 and 0.99 over ocean and still reaches 0.87 […]

  Restated for clarification.

  "The correlation regarding AOT in that scene is 0.97 (670 nm) and 0.99 (865 nm) over ocean and reaches 0.87 (670 nm) over land, [...]" (L425-426)

- L422 "relatively bright" - please avoid this kind of statement and be precise

  Thank you. Removed "relatively" from the sentence.

---

## Author Comment (AC2)

**Reply to Anonymous Referee #2**

We thank the reviewer for their helpful and constructive comments. Please find our answers (black text) to the comments (blue text) in-line below. Respective changes are indicated in the revised manuscript in blue and are stated here in addition when a reviewer comment lead to a modification of the manuscript along with the updated line number, where necessary.

The upcoming EarthCARE mission will include measurements of M-AOT, or column aerosol optical depth at 670 nm (over land and ocean) and 865 nm (over ocean). The study presents the algorithm for these retrievals and results using simulated EarthCARE data and the MODIS L1 reflectances, comparing this output to existing AOD measurements from AERONET and MAN and to the MODIS aerosol products. It is strange that for both the synthetic test scenes and the MSI algorithm applied to MODIS L1 reflectances, the performance is dramatically better over ocean than over land; this raises questions about its surface reflectance estimation. The comparisons to MODIS and AERONET data need clearer justification.

Thank you for your comment.

Regarding your first point "the performance is dramatically better over ocean than over land": We were actually not surprised about the worse performance over land than over water due to the difficult task of separating the surface and the atmospheric contribution in the TOA signal as stated several times throughout the manuscript. Nonetheless, we additionally clarified in the discussion section by adding:

"In general and as common for near-real time aerosol products over land based on similar passive measurements, the performance of the M-AOT product is worse over land than over ocean. This is due to the very strict surface and aerosol composition assumptions made and due to the lack of higher accuracy surface information available at operational run-time. Nonetheless, a correlation of 0.77 (AERONET) and 0.87 (MODIS MYD04) could be reached using MODIS Level 1 data in the M-AOT algorithm." (L466-470)

Further, we added justifications for comparisons to AERONET and MODIS in the manuscript in L412 (MODIS), L438-440 (AERONET).

My other specific comments are below:

- Line 16. "Aerosols have a special role in the overall context" of what? Consider rephrasing. Rephrased to:

  "Aerosols have a special role in the overall context of radiative interactions in the atmosphere [...]" (L16-17)

- Line 21. Another unclear use of "context". Does this mean the lidar data needs to be combined with MSI to obtain geolocation?
  Since the overall EarthCARE concept, is part of a different publication in this special issue, as already referenced in the manuscript, we will not go into more detail in the manuscript itself than we already have. However, to answer your question:
  No. Here it is referred to the fact that, by measurement nature, a space-based lidar is capable to provide vertical information, but only along track (i.e. no across track or horizontal

information) while a space-based imager is able to provide horizontal information, but only for one observed column as a whole (i.e. no vertical information). This in turn means that one could gain additional knowledge from combining these two measurements of one observed location. It also means that you hypothetically could see e.g. the vertical location of a smoke plume in the lidar measurements. However, the imager measurements might offer you the possibility to add information to the lidar knowledge by e.g. hinting at the actual source region of that plume a few kilometers away. Hence, MSI data will be able to add information for that hypothetical case.

- Lines 82-83. What is the spatial resolution of one MSI pixel? Measurements of aerosols at cloud edges (and of aerosols in or near dust or smoke plumes thick enough to be mistaken for clouds) are valuable in themselves, and this seems like it would greatly reduce spatial coverage.
  The actual statement about the spatial resolution of MSI is already available in L58 of the manuscript.
  The various radiative effects close to and at cloud edges cannot be considered in such a simple aerosol retrieval that is supposed to deliver the product in near-real time. This limitation is shared with all other passive imager based aerosol retrievals of that kind. Additionally, we added a clarification:
  "[...] are flagged as well during that step in order to avoid [...] and other three dimensional radiative transfer effects. [...] 3 pixel, which is corresponding to 1.5 km." (L95-96)

- Lines 232-334. How often would the land cover type map be updated to account for land use change?
  Currently, it is not foreseen to be updated. However, we could imagine that the need for an update arises during future developments.

- Fig. 3 and Fig. 4. If there's a way to generate RGB images for these synthetic data test scenes, they would be helpful to orient the reader.

  RGB images are not directly available due to the band setting of MSI (i.e. no channel availability in the blue or green).

- Meanwhile, the skill seems very low for the land retrieval, but both scenes are mostly ocean; it's hard to tell how much of this is limited sampling.

  We agree that the algorithm performs worse over land than ocean. However, this is as expected due to the complex task of disentangling surface and atmospheric contribution to the TOA signal.

- Line 357. A reference placeholder (?) has been left unfilled.

  Thank you. Updated. (L394)

- Lines 374-375. The references are for the MODIS Dark Target algorithm, but the MODIS L2 aerosol file includes retrievals from Dark Target, the Deep Blue algorithm, and a combination DTDB. Make sure to specify which is meant.

  Updated. "Dark target" is specifically stated now. (L411)

-  Unless the MODIS comparison is meant to use the Deep Blue or DTDB retrievals exclusively, the L2 product reports Dark Target AOD at 660 nm (land and ocean) and 860 nm (ocean) directly. Inferring the values from the Ångström exponent and the 550 nm retrieval instead risks losing precision.

  Agreed. However, the M-AOT product reports AOT at 670 nm and 865 nm. In order to compare AOT at the same wavelength, MODIS dark target AOT was extrapolated to these wavelength. The loss in precision is expected to be negligible compared to other potential sources of inconsistencies such as different assumptions about e.g. aerosol types or surface parameterization.

  Nonetheless, this pre-launch exercise has been executed in order to give an indication of the expected performance of the EarthCARE MSI stand-alone aerosol retrieval.

-  The MODIS Ångström exponents are calculated for ocean only by the Dark Target retrieval and for land only by Deep Blue, so the underlying algorithms being compared here are not equivalent. Also, double check these wavelengths.

  Even though it is stated that the transfer (or "extrapolation" in the updated manuscript) has been executed via the Ångström parameter, it was not stated that this is the "official" MODIS reported value but rather that it is based (calculated) on the official AOTs. Hence, only the dark target AOT values have been used to our best knowledge. The wavelength have been double checked and are based on the reported "long_names" in the MODIS HDF product(s).

- Fig. 6. The corresponding MODIS RGB image would be helpful here, too.

  We omit the RGB image here too in order to keep the general structure of both verification sections as equal as possible and kindly refer you to publically and freely available MODIS RGBs using for instance NASA worldview https://worldview.earthdata.nasa.gov/ (last visited 28 Apr 2023)

- Fig. 8. Any idea why these scatterplots form branches at higher AOD values? It's still odd that the correlation for land is so much lower than for ocean.

  Branching for higher AOD:
  As already stated in the manuscript (L434-435), this is mainly attributed to different assumptions in the MODIS and M-AOT retrieval, in particular aerosol type choice.
  We further added: "The resulting AOT differences exist for both, low and higher AOT. However, they become amplified for higher AOTs." (L435-436)

  Land vs ocean:
  We are not surprised that the correlation is worse over land than over ocean. As stated in subsection 2.3.4, the contribution of the surface to the TOA signal is stronger over land than over ocean. Hence, it is much more difficult to separate the atmosphere and surface contribution to the TOA signal from another. This in turn makes it much more difficult to precisely estimate the AOT over land.

-  5 km is a tighter spatial radius than is commonly used for satellite/AERONET matches, and this does not necessarily improve representativeness. How were the match criteria chosen?

  This value has been chosen according to a common amount of pixel used and stated for

example in Concha et al 2021. There, for ocean match-up validation studies for example, a value between 3x3 and 7x7 pixel is used. Translated to the MODIS resolution used in M-AOT of 1 km times 1 km, our chosen value of 5 km lies right in the middle.

*Javier A. Concha, Marco Bracaglia, Vittorio E. Brando, Assessing the influence of different validation protocols on Ocean Colour match-up analyses, Remote Sensing of Environment, Volume 259,2021, 112415, ISSN 0034-4257, https://doi.org/10.1016/j.rse.2021.112415.*

- Line 414. Are there plans to report a 670 nm/865 nm Ångström exponent as part of the MSI product?

Yes, it is planned to be part of the MSI product.